# Acetylcholine is released in the basolateral amygdala in response to predictors of reward and enhances the learning of cue-reward contingency

Richard B Crouse[1,2], Kristen Kim[1,2], Hannah M Batchelor[1,2], Eric M Girardi[1],
Rufina Kamaletdinova[1,3], Justin Chan[1], Prithviraj Rajebhosale[4,5],
Steven T Pittenger[1], Lorna W Role[5], David A Talmage[6], Miao Jing[7], Yulong Li[8,9,10],
Xiao-Bing Gao[11], Yann S Mineur[1], Marina R Picciotto[1,2]*

[1]Department of Psychiatry, Yale University, New Haven, United States; [2]Yale Interdepartmental Neuroscience Program, New Haven, United States; [3]City University of New York, Hunter College, New York, United States; [4]Program in Neuroscience, Stony Brook University, New York, United States; [5]National Institute of Neurological Disorders and Stroke (NINDS), Bethesda, United States; [6]National Institute of Mental Health (NIMH), Bethesda, United States; [7]Chinese Institute for Brain Research (CIBR), Beijing, China; [8]State Key Laboratory of Membrane Biology, Peking University School of Life Sciences, Beijing, China; [9]PKU-IDG/McGovern Institute for Brain Research, Beijing, China; [10]Peking-Tsinghua Center for Life Sciences, Academy for Advanced Interdisciplinary Studies, Peking University, Beijing, China; [11]Section of Comparative Medicine, Yale University School of Medicine, New Haven, United States

*For correspondence:
marina.picciotto@yale.edu

Competing interests: The authors declare that no competing interests exist.

**Abstract** The basolateral amygdala (BLA) is critical for associating initially neutral cues with appetitive and aversive stimuli and receives dense neuromodulatory acetylcholine (ACh) projections. We measured BLA ACh signaling and activity of neurons expressing CaMKIIα (a marker for glutamatergic principal cells) in mice during cue-reward learning using a fluorescent ACh sensor and calcium indicators. We found that ACh levels and nucleus basalis of Meynert (NBM) cholinergic terminal activity in the BLA (NBM-BLA) increased sharply in response to reward-related events and shifted as mice learned the cue-reward contingency. BLA CaMKIIα neuron activity followed reward retrieval and moved to the reward-predictive cue after task acquisition. Optical stimulation of cholinergic NBM-BLA terminal fibers led to a quicker acquisition of the cue-reward contingency. These results indicate BLA ACh signaling carries important information about salient events in cue-reward learning and provides a framework for understanding how ACh signaling contributes to shaping BLA responses to emotional stimuli.

## Introduction

Learning how environmental stimuli predict the availability of food and other natural rewards is critical for survival. The basolateral amygdala (BLA) is a brain area necessary for associating cues with both positive and negative valence outcomes (*Baxter and Murray, 2002*; *Janak and Tye, 2015*; *LeDoux et al., 1990*). A recent study has shown that genetically distinct subsets of BLA principal neurons encode the appetitive and aversive value of stimuli (*Kim et al., 2016b*). This encoding

involves the interplay between principal neurons, interneurons, and incoming terminal fibers, all of which need to be tightly regulated to function efficiently.

The neuromodulator acetylcholine (ACh) is released throughout the brain and can control neuronal activity via a wide range of mechanisms. ACh signals through two families of receptors (nicotinic, nAChRs and muscarinic, mAChRs) that are differentially expressed on BLA neurons as well as their afferents (*Picciotto et al., 2012*). ACh signals through these receptors to increase signal-to-noise ratios and modify synaptic transmission and plasticity in circuits involved in learning new contingencies (*Picciotto et al., 2012*), especially in areas that receive dense cholinergic input, like the BLA (*Woolf, 1991*; *Zaborszky et al., 2012*). The effect of ACh signaling can differ depending on the receptor, as metabotropic mAChRs work on a slower timescale than the rapid, ionotropic nAChRs (*Gu and Yakel, 2011*; *Picciotto et al., 2012*). The overall impact of ACh signaling on the BLA is likely quite heterogeneous as mAChRs are coupled to both inhibitory and excitatory signaling cascades and nAChRs are found on both glutamatergic and GABAergic BLA neurons (*Picciotto et al., 2012*).

The basal forebrain complex is a primary source of ACh input to the BLA. In particular, the nucleus basalis of Meynert (NBM) sends dense cholinergic projections to the BLA (*Woolf, 1991*; *Zaborszky et al., 2012*). Optical stimulation of BLA-projecting cholinergic terminal fibers (NBM-BLA) during fear conditioning is sufficient to strengthen fear memories (*Jiang et al., 2016*) and may support appetitive behavior (*Aitta-aho et al., 2018*). Cholinergic NBM neurons increase their firing in response to both rewarding and aversive unconditioned stimuli (*Hangya et al., 2015*). Cholinergic signaling in the medial prefrontal cortex and visual cortex has been linked to cue detection (*Parikh et al., 2007*) and reward timing (*Chubykin et al., 2013*; *Liu et al., 2015*), respectively. A recent study has also demonstrated that NBM cells fire in response to a conditioned stimulus during trace fear conditioning, indicating that ACh signaling may be involved in learning about cues that predict salient outcomes (*Guo et al., 2019*).

We hypothesized that ACh signaling in the BLA is a critical neuromodulatory signal that responds to both unconditioned stimuli and cues that gain salience, thereby coordinating activity in circuits necessary for learning cue-reward contingencies. To test this hypothesis, we measured relative levels of BLA ACh (ACh signaling), cholinergic NBM-BLA terminal fiber activity (BLA ACh signal origin), and the activity of BLA principal neurons (BLA output) across all phases of learning in an appetitive operant learning task to evaluate how BLA output and ACh signaling are related to behavioral performance in this paradigm. We then optically stimulated cholinergic NBM fibers locally in the BLA, while mice learned to nose poke in response to an auditory cue to receive a food reward to determine if accelerating the increase in ACh signaling that occurs as mice learn the task would enhance performance. We also pharmacologically blocked different ACh receptors during the learning task to determine the subtypes involved, and varied the timing of optical stimulation of cholinergic NBM-BLA terminal fibers to determine whether ACh release time-locked with the reward-predictive cue is necessary for the improvement of the task performance. These studies provide a novel framework for understanding how NBM ACh signaling in the BLA is recruited during the perception of novel stimuli and how it contributes to linking previously neutral cues to predictions about future salient outcomes.

## Results

### Acetylcholine release in the BLA occurs at salient points in the cue-reward learning task and shifts as mice learn the cue-reward contingency

The BLA is critical for learning that previously neutral cues can predict future punishments or rewards and for assigning valence to those cues (*Baxter and Murray, 2002*; *Janak and Tye, 2015*). The BLA receives dense cholinergic input (*Woolf, 1991*; *Zaborszky et al., 2012*) and we speculated that, since ACh signaling is involved in both attention and several types of learning (*Picciotto et al., 2012*), it could be essential for learning about cues that predict salient events, such as reward delivery. Based on data showing that ACh neurons fire in response to unexpected or salient events (*Hangya et al., 2015*), we also hypothesized that ACh release might vary as mice learn a cue-reward contingency. Therefore, we designed a cue-reward learning task in which food-restricted mice were

trained to perform a nose poke when signaled by a cue (tone) to receive a palatable reward (Ensure) on a 30 s variable intertrial interval (ITI; *Figure 1A–D*). We injected adeno-associated virus (AAV) carrying an improved version of the fluorescent ACh sensor GRAB~ACh3.0~ (ACh3.0; *Jing et al., 2018*; *Jing et al., 2019*) construct into the BLA of mice and implanted an optical fiber above the BLA to record ACh signaling during the cue-reward learning task (*Figure 2A* + *Figure 2—figure supplement 1A*).

During the Pre-Training phase of the task, mice received reward and receptacle light presentation for performing a nose poke in the active port during tone presentation (*Figure 1C*, purple active nose poke coincident with tone) but there was no consequence for an incorrect nose poke (*Figure 1C*, red active nose poke not coincident with tone). Animals quickly learned to make a high number of responses over the course of each Pre-Training session. In this paradigm, mice obtained most available rewards by day 5 of Pre-Training (*Figure 2B*, blue shaded region). However, this phase of training did not promote learning of the cue-reward contingency, (i.e. that they should only nose poke during tone presentation) seen by the high number of incorrect nose pokes (*Figure 2—figure supplement 2A*, blue shaded region). Mice performed roughly eightfold more incorrect nose pokes than correct nose pokes, suggesting that mice were not attending to the task contingency. The Training phase of the task was identical to Pre-Training except incorrect nose pokes resulted in a 5 s timeout, during which the house light was illuminated, that concluded with a restarting of the ITI timer (*Figure 1D*, red active nose poke not coincident with tone). On day 1 of the Training phase, all animals earned fewer rewards (*Figure 2B*, pink shading) and, while still high, incorrect nose pokes dropped (*Figure 2—figure supplement 2A*, pink shading). Animals that did not meet acquisition criterion by day 9 (defined as consistently earning 20 or more rewards per session, *Figure 2B*, white horizontal line) were moved to a 20 s variable ITI to promote responding (*Figure 2B*, pink shading day 10). Following the change in ITI, mice acquired the cue-reward behavior at different rates. After

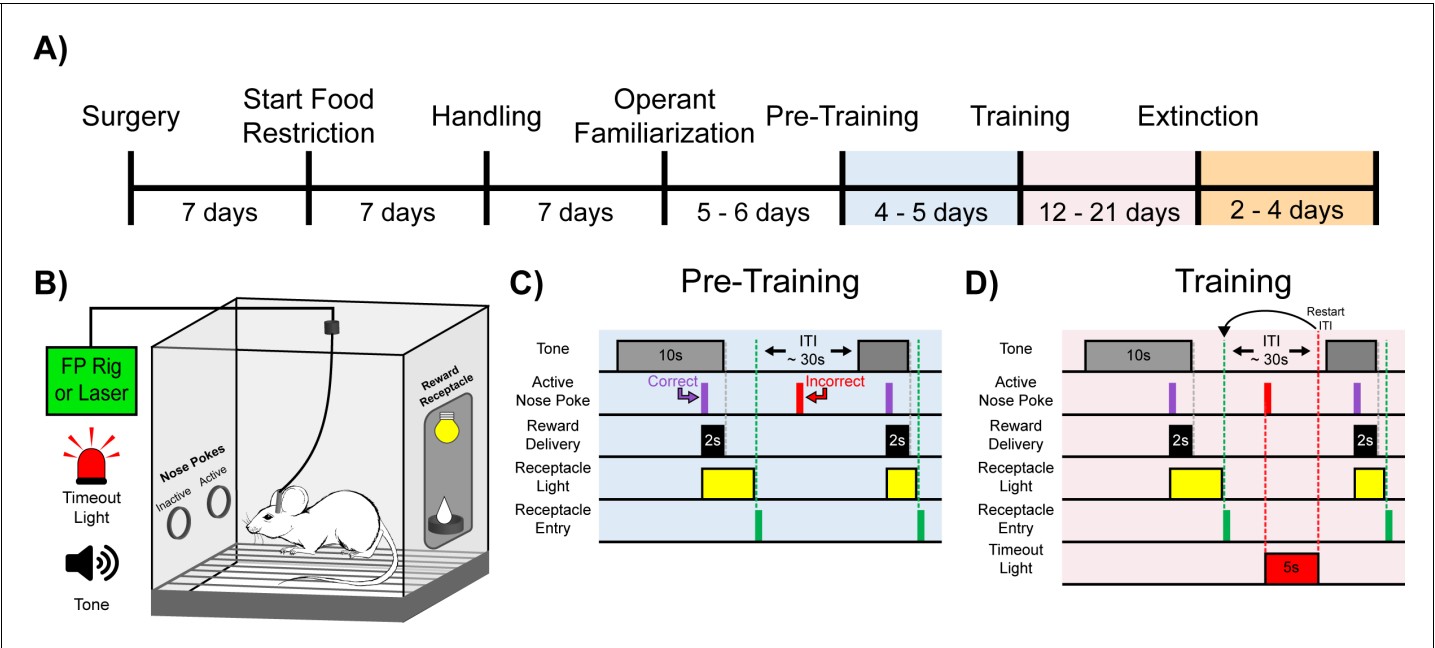

**Figure 1.** Experimental timeline and cue-reward learning paradigm. (**A**) Experimental timeline. Mice began food restriction 7 d after surgery and were maintained at 85% free-feeding body weight for the duration of the experiment. After 7 d of handling, 5–6 d of operant familiarization prepared the mice for the cue-reward learning task (Pre-Training through Extinction). (**B**) Behavioral chamber setup. Mice were placed in modular test chambers that included two nose poke ports on the left wall (Active and Inactive) and the Reward Receptacle on the right wall. A tone generator and timeout light were placed outside the modular test chamber. For fiber photometry (FP) and optical stimulation (Laser) experiments, mice were tethered to a patch cord(s). (**C–D**) Details of the Cue-Reward Learning Paradigm. (**C**) In Pre-Training, an auditory tone was presented on a variable interval 30 schedule (VI30), during which an active nose poke yielded Ensure reward delivery but there was no consequence for incorrect nose pokes (active nose pokes not during tone). (**D**) Training was identical to Pre-Training, except incorrect nose pokes resulted in a 5 s timeout, signaled by timeout light illumination, followed by a restarting of the intertrial interval (ITI).

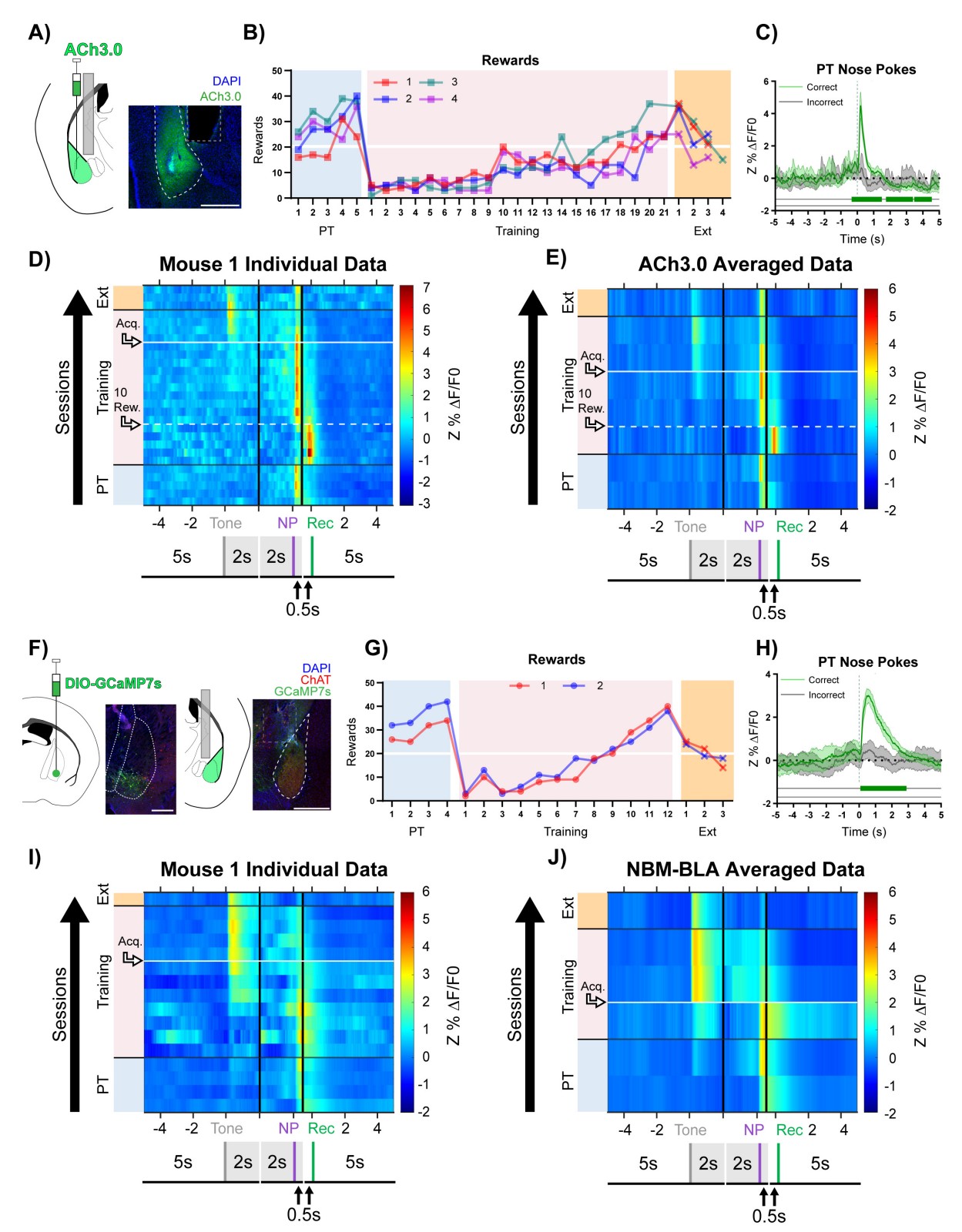

**Figure 2.** Basolateral amygdala (BLA) ACh signaling aligns with salient events during reward learning. (**A**) Diagram and example of injection and fiber placement sites in the BLA for recording from mice expressing a fluorescent acetylcholine sensor (ACh3.0). Left: Diagram of BLA ACh3.0 injection and fiber tip placement. Right: Representative coronal brain slice with fiber tip and ACh3.0 expression. Blue: DAPI, Green: ACh3.0. White dashed line: BLA outline. Gray dashed rectangle: fiber track. Scale = 500 μm. Individual fiber placements are shown in *Figure 2—figure supplement 1A*. (**B**) Behavioral
*Figure 2 continued on next page*

*Figure 2 continued*

responses of mice expressing ACh3.0 in BLA. Individual mice acquired the task at different rates as measured by rewards earned. Horizontal white line: acquisition threshold, when a mouse began to earn 20 rewards consistently in Training. Incorrect nose pokes shown in *Figure 2—figure supplement 2A*. Pre-Training (PT): blue shaded area, Training: pink shaded area, Extinction (Ext): orange shaded area. (C) Fluorescence traces from BLA of ACh3.0-expressing mouse. A significant increase in fluorescence representing BLA ACh release consistently coincided with correct (green line) but not incorrect (gray line) nose pokes on the last day of PT (data are shown from Mouse 1). The mean Z-scored precent ΔF/F0 (Z%ΔF/F0) overlaid on bootstrapped 99% confidence intervals (99% bCIs). Shaded significance bars under traces represent time points where 99% bCIs do not contain 0 for at least 0.5 s. Correct: n = 24; downsampled incorrect: n = 24 of 58. Traces of signal and reference channels (%ΔF/F0) during nose pokes are shown in *Figure 2—figure supplement 1B–C*. Incorrect nose pokes on the last day of PT versus Training Day 1 shown in *Figure 2—figure supplement 2B*. (D) Heatmap of BLA ACh signaling in Mouse 1 across all training phases, aligned to tone onset (Tone), correct nose poke (NP), and receptacle entry (Rec). Each row is the average of rewarded trials across a training session. White dashed horizontal line: first Training day earning 10 rewards. Horizontal white line: acquisition threshold, when a mouse began to earn 20 rewards consistently in Training. Black horizontal lines: divisions between training phases. Black vertical lines: divisions between breaks in time to allow for variable latencies in tone onset, correct nose poke, and receptacle entry (reward retrieval). The bCI plot for Mouse 1 is in *Figure 2—figure supplement 1G*. Individual heatmaps for mice 2–4 in *Figure 2—figure supplement 1D–F*. Incorrect nose pokes heatmaps for individual mice shown in *Figure 2—figure supplement 2C–F*. (E) Heatmap of BLA ACh signaling averaged across mice. Signal aligned as in (D) with a selection of data from key days in the behavioral paradigm shown. From bottom to top: PT Day 1, PT Day 5, Early Training Day, First Training day earning 10 rewards (white dashed horizontal line), Mid Training Day, Acquisition Day (white horizontal line), Last Training Day, Last Extinction Day. The bCI plot for cohort averaged data is in *Figure 2—figure supplement 1H*. Incorrect nose poke heatmap and bCI plot averaged across mice shown in *Figure 2—figure supplement 2G–H*. (F) Diagram and example of Nucleus Basalis of Mynert (NBM)-BLA terminal fiber recordings. Left: DIO-GCaMP7s was injected in the NBM of ChAT-IRES-Cre mice, individual injection sites are shown in *Figure 2—figure supplement 5A*. Representative coronal brain slice showing GCaMP7s expression. White dashed lines: internal capsule and globus pallidus outlines. Blue: DAPI, Green: GCaMP7s, Red: ChAT. Scale = 500 µm; separate channels shown in *Figure 2—figure supplement 5B*. Right: An optical fiber was implanted above the ipsilateral BLA, individual fiber placements are shown in *Figure 2—figure supplement 5A*. Representative coronal brain slice showing GCaMP7 expression and fiber tip placement. White dashed line: BLA outline. Gray dashed rectangle: fiber tract. Blue: DAPI, Green: GCaMP7s, Red: ChAT. Scale = 500 µm; separate channels shown in *Figure 2—figure supplement 5C*. (G) Behavioral responses of mice during NBM-BLA terminal fiber recordings. White horizontal line: acquisition threshold, when a mouse began to earn 20 rewards consistently in Training. Incorrect nose pokes shown in *Figure 2—figure supplement 6A*. (H) NBM-BLA terminal fiber activity is similar to ACh3.0 recordings. NBM-BLA terminal fiber activity significantly increased with correct (green line) but not incorrect (gray line) nose pokes on the last day of PT (data shown for Mouse 1). Mean Z%ΔF/F0 overlaid on bootstrapped 99% confidence intervals (99% bCIs). Shaded significance bars under traces represent time points where 99% bCIs do not contain 0 for at least 0.5 s. Correct: n = 42; downsampled incorrect: n = 42 of 101. Signal and reference channels (%ΔF/F0) during nose pokes are shown in *Figure 2—figure supplement 5D–E*. Incorrect nose pokes on the last day of PT versus Training Day 1 shown in *Figure 2—figure supplement 6B*. See *Figure 2—figure supplement 9A–H* for simultaneous ACh3.0 and NBM-BLA terminal fiber recordings. (I) Heatmap of NBM-BLA terminal fiber activity in Mouse 1 across all training phases, as in (D-E). Blanks in the heatmaps indicate time bins added for alignment. bCI plot for Mouse 1 in *Figure 2—figure supplement 7F*. Mouse 2 individual heatmap shown in *Figure 2—figure supplement 5F*. Incorrect nose pokes heatmaps for individual mice shown in *Figure 2—figure supplement 6C–D*. (J) Heatmap of NBM-BLA terminal fiber activity averaged across mice. Signal aligned as in (D-E, I) with a selection of key days shown, from bottom to top: PT Day 1, PT Day 4, Early Training, Acquisition Day (white horizontal line), Last Training Day, Last Extinction Day. The bCI plot for cohort averaged data in *Figure 2—figure supplement 7G*. Incorrect nose poke heatmap and bCI plot averaged across mice shown in *Figure 2—figure supplement 6E* + *Figure 2—figure supplement 8E*.

The online version of this article includes the following figure supplement(s) for figure 2:

**Figure supplement 1.** BLA ACh3.0 recording.
**Figure supplement 2.** BLA ACh3.0 recording: incorrect nose pokes.
**Figure supplement 3.** BLA ACh3.0 recording replicate.
**Figure supplement 4.** BLA ACh3.0 recording replicate: incorrect nose pokes.
**Figure supplement 5.** NBM-BLA GCaMP7s recording in cholinergic terminal fibers.
**Figure supplement 6.** NBM-BLA GCaMP7s recording in cholinergic terminal fibers: incorrect nose pokes.
**Figure supplement 7.** NBM-BLA GCaMP7s recording in cholinergic terminal fibers replicate.
**Figure supplement 8.** NBM-BLA GCaMP7s recording in cholinergic terminal fibers replicate: incorrect nose pokes.
**Figure supplement 9.** Simultaneous BLA ACh3.0 + GCaMP7s recording in NBM-BLA cholinergic terminal fibers.

acquisition, animals were switched to Extinction training in which correct nose pokes did not result in reward delivery, and all mice decreased nose poke responding (*Figure 2B* + *Figure 2—figure supplement 2A*, orange shading).

During Pre-Training, when there were high numbers of both correct and incorrect nose pokes, there was a large increase in ACh release following correct nose pokes, which were followed by reward delivery and receptacle light, but not incorrect nose pokes (*Figure 2C* + *Figure 2—figure supplement 1B–C*). We used bootstrapped confidence intervals (bCIs) to determine when transients were statistically significant (bCI did not contain the null of 0 [*Jean-Richard-Dit-Bressel et al., 2020*]). Correct, but not incorrect, nose poke trials consistently showed a sustained, significant

increase in fluorescence close to the time of nose poke onset (*Figure 2C*). We also observed a significant decrease in fluorescence for most mice around 2–4 s after correct nose poke, which corresponds to the time of reward retrieval.

ACh release occurred in response to different events as mice learned the task (data for individual mice are shown in *Figure 2D* + *Figure 2—figure supplement 1D–G* and averaged data across all mice at key time points in the task is shown in *Figure 2E* + *Figure 2—figure supplement 1H*). During Pre-Training rewarded trials, the highest levels of ACh release occurred close to the time of correct nose pokes (NP), with a smaller peak at the time of reward retrieval (entry into the reward receptacle, Rec). As Training began, the ACh release during reward trials shifted dramatically toward the time of reward retrieval, likely because the animals were learning that many nose pokes did not result in reward delivery. Incorrect nose pokes that triggered a timeout were also followed by a modest but non-significant increase in BLA ACh levels (*Figure 2—figure supplement 2B–H*). As mice began to learn the contingency (*Figure 2E* + *Figure 2—figure supplement 1H*, 10 rewards), the peak ACh release during rewarded trials shifted back to the time of the correct nose poke response. As animals approached the acquisition criterion (*Figure 2E* + *Figure 2—figure supplement 1H*, Acq.), ACh level significantly increased at the time of the tone, suggesting that as animals learned the cue-reward contingency, the tone became a more salient event. At this time point, there was still a peak at the time of reward, but its magnitude was diminished. After task acquisition, the increase in ACh following correct nose pokes remained but was diminished, and incorrect nose pokes did not elicit apparent ACh release (*Figure 2—figure supplement 2C–H*, Acq.). During Extinction, ACh release to tone onset diminished. We replicated this experiment in an independent cohort of mice and found similar results (*Figure 2—figure supplements 3–4*). Mice in this replicate cohort learned in a similar fashion (*Figure 2—figure supplement 3B* + 4A) but met the acquisition criteria faster than initial mice because aspects of the behavioral setup were optimized (3D printed wall extensions) to allow the imaging apparatus to be used inside sound attenuating chambers (see Materials and methods section). One difference observed in this group that learned the task more rapidly, was small magnitude, but significant, increases in BLA ACh release following tone onset late in Pre-Training (*Figure 2—figure supplement 3C–I*). As behavioral performance during the Training phase increased, ACh release to tone onset became more pronounced, as in the initial cohort.

In order to determine the source of the ACh released in the BLA during cue-reward learning, we recorded calcium dynamics as a measure of cell activity of ChAT$^+$ NBM terminal fibers in the BLA (NBM-BLA), since the NBM is a major source of cholinergic input to the BLA (*Jiang et al., 2016*; *Woolf, 1991*; *Zaborszky et al., 2012*). We injected AAV carrying a Cre-recombinase-dependent, genetically-encoded calcium indicator (DIO-GCaMP7s) into the NBM of ChAT-IRES-Cre mice and implanted an optical fiber above the ipsilateral BLA (*Figure 2F* + *Figure 2—figure supplement 5A–C*). As with the ACh3.0 sensor, there was a significant increase in NBM-BLA cholinergic terminal activity following correct, but not incorrect, nose pokes (*Figure 2H* + *Figure 2—figure supplement 5D–E*). NBM-BLA cholinergic terminal activity evolved across phases of the reward learning task as was seen for ACh levels in the BLA (data for each mouse shown in *Figure 2I* + *Figure 2—figure supplement 5F–G*, averaged across all mice at key time points in the task shown in *Figure 2J* + *Figure 2—figure supplement 7G*). Strikingly, NBM-BLA cholinergic terminal activity most closely followed correct nose pokes in Pre-Training and shifted primarily to tone onset as mice learned the contingency during Training. As in the replication cohort for the ACh sensor, small magnitude, but significant, increases in terminal activity were observed following tone onset late in Pre-Training (*Figure 2J* + *Figure 2—figure supplement 7G*). Incorrect nose pokes that resulted in a timeout in Training sessions were followed by a modest increase in NBM-BLA cholinergic terminal activity before task acquisition (*Figure 2—figure supplement 6B–E*). During Extinction, activity of NBM-BLA terminals following tone onset diminished. These findings were replicated in an independent cohort of mice, which we combined for across-mouse statistical analyses (*Figure 2—figure supplements 7–8*).

In order to record NBM-BLA cholinergic terminal activity and BLA ACh levels simultaneously in the same mouse, we injected AAV carrying a construct for Cre-recombinase dependent red-shifted genetically-encoded calcium indicator (DIO-jRCaMP1b) into the NBM of ChAT-IRES-Cre mice, ACh3.0 sensor into the ipsilateral BLA, and implanted a fiber above the BLA (*Figure 2—figure supplement 9A–E*, mouse 1). DIO-jRCaMP1b was also injected into the NBM of a wild-type littermate so Cre-mediated recombination would not occur to control for any crosstalk between the ACh3.0

and jRCaMP1b channels. While this was only a single animal and proof of principle for future studies, we found that NBM-BLA cholinergic terminal activity coincided with ACh levels (*Figure 2—figure supplement 9F–G*). Importantly, this relationship between ACh release and NBM-BLA terminal fiber activity was not explained by signal crosstalk (*Figure 2—figure supplement 9H–I*), further indicating that the BLA ACh measured comes at least in part from the NBM.

## BLA principal neurons respond to reward availability and follows cue-reward learning

Glutamatergic principal cells are the primary output neurons of the BLA (*Janak and Tye, 2015*), and their firing is modulated by NBM-BLA cholinergic signaling (*Jiang et al., 2016*; *Unal et al., 2015*). BLA principal neurons can increase their firing in response to cues as animals learn cue-reward contingencies (*Sanghera et al., 1979*; *Schoenbaum et al., 1998*; *Tye and Janak, 2007*). Calcium/calmodulin-dependent protein kinase (CaMKII) has been shown to be a marker for glutamatergic BLA principal cells (*Butler et al., 2011*; *Felix-Ortiz and Tye, 2014*; *McDonald, 1992*; *Tye et al., 2011*). To determine whether ACh modulates principal neuron activity during cue-reward learning, we injected AAV carrying a Cre-recombinase dependent genetically encoded calcium indicator (DIO-GCaMP6s) into the BLA of CaMKIIα-Cre mice to record BLA principal cell activity during the learning task (*Figure 3A* + *Figure 3—figure supplement 1A*). As was seen for BLA ACh levels, there was a significant increase in BLA CaMKIIα cell activity following correct and a modest decrease in activity following incorrect nose pokes on the last day of Pre-Training (*Figure 3B*). However, the activity peaked later after the correct nose poke response (~2.5 s) compared to the ACh3.0 signal (~0.5 s) and appeared to align more tightly with reward retrieval (*Figure 3—figure supplement 1B*). As mice learned the task (*Figure 3C* + *Figure 3—figure supplement 2A*), BLA CaMKIIα cell activity increased first in response to reward and, after acquisition of the task, to the reward-predictive cue (individual data for each mouse shown in *Figure 3D* + *Figure 3—figure supplement 1E–F*, and averaged data across all mice at key time points in task is shown in *Figure 3E* + *Figure 3—figure supplement 1G–H*).

During Pre-Training, the highest levels of BLA CaMKIIα cell activity followed reward retrieval. In addition, during the first few days of Training, BLA CaMKIIα cell activity after reward retrieval was higher than it was during Pre-Training, and the magnitude of response decreased as mice learned the contingency and earned more rewards, ultimately reaching similar intensity to that observed during Pre-Training. Concurrently, as mice approached acquisition of the task (*Figure 3C*, white horizontal line), BLA CaMKIIα cell activity significantly increased in response to tone onset (*Figure 3D-E* + *Figure 3—figure supplement 1E–H*, Acq.), suggesting that the recruitment of BLA CaMKIIα cell activity likely reflects the association of the cue with a salient outcome (*Lutas et al., 2019*; *Sengupta et al., 2018*). Incorrect nose pokes that triggered a timeout did not elicit a different response in CaMKIIα cell activity compared to before timeouts were incorporated (*Figure 3—figure supplement 2B–G*). In an independent cohort of mice, those with more posterior fiber tip placements (mice 4 + 7) replicated the primary findings (*Figure 3—figure supplements 3–4*).

## Stimulation of cholinergic terminals in BLA improves cue-reward learning

Since ACh released by NBM-BLA terminals during Training shifted to tone onset during acquisition of cue-reward learning (*Figure 2E,J*), we hypothesized that ACh may potentiate learning the cue-reward contingency. We, therefore, tested whether increasing ACh release in BLA during learning could alter cue-reward learning by injecting AAV carrying a Cre-recombinase-dependent channelrhodopsin-EYFP (AAV-DIO-ChR2-EYFP) construct bilaterally into the NBM of ChAT-IRES-Cre transgenic mice and placing fibers over the BLAs to optically stimulate cholinergic terminals originating from the NBM selectively (*Figure 4A* + *Figure 4—figure supplement 1*). Optical control over ChAT[+] NBM cells was verified by ex vivo slice recordings, depolarizations followed light flashes and clear action potentials were observed ex vivo (*Figure 4B* + *Figure 4—figure supplement 2*). After operant familiarization, ChAT[+] NBM-BLA terminals were stimulated via bilateral optical fibers (2 s, 20 Hz, 25 ms pulses) triggered by a correct nose poke throughout both Pre-Training (*Figure 4C*) and Training (*Figure 4D*). Stimulation usually occurred during at least a portion of all three components of a

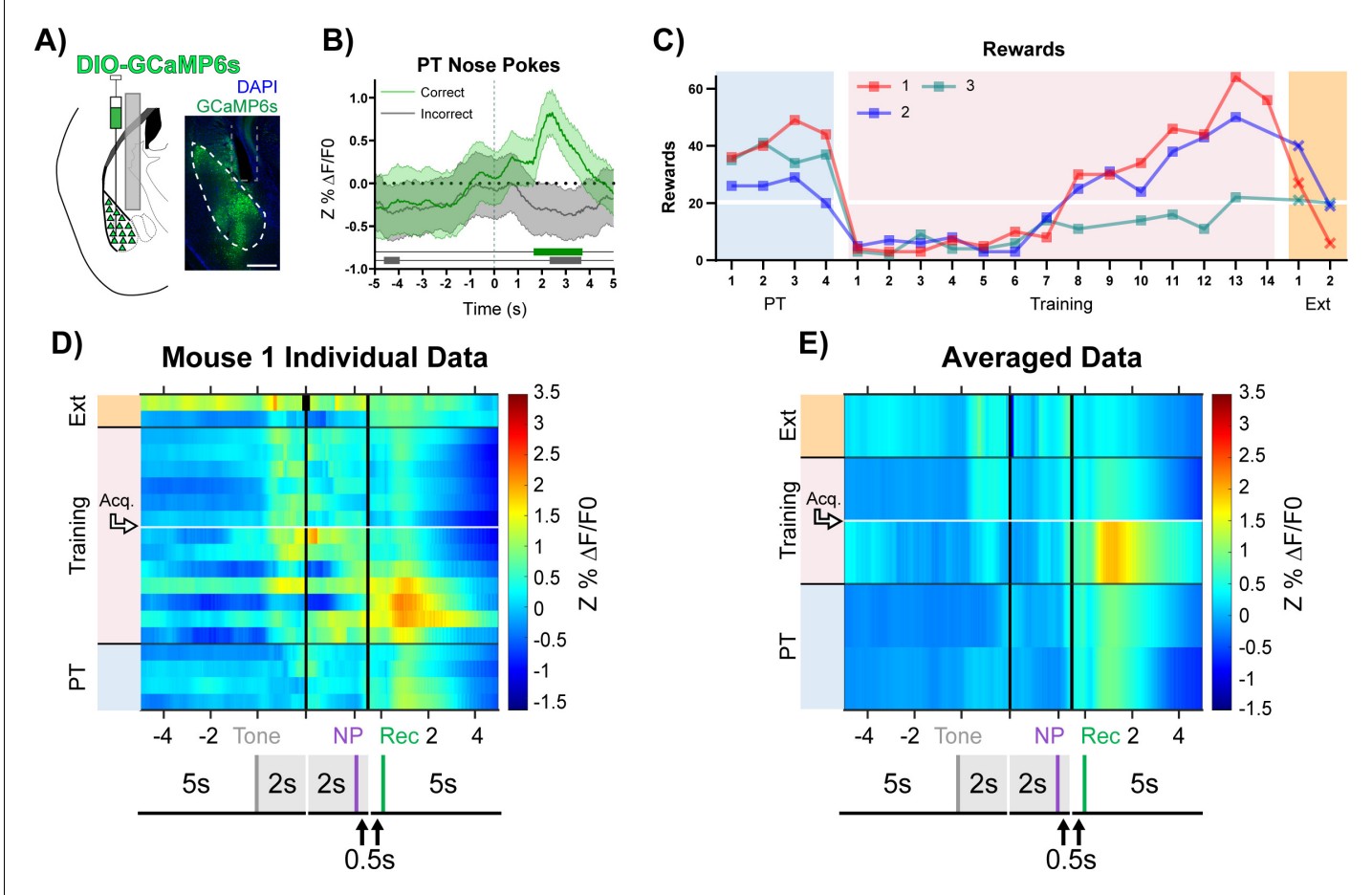

**Figure 3.** BLA CaMKIIα neuron activity aligns to reward retrieval and cue-reward learning. (**A**) Diagram and example of injection and fiber placement sites in the BLA for recording from CaMKIIα-Cre mice expressing a fluorescent calcium indicator (DIO-GCaMP6s). Left: Diagram of injection and fiber placement. Right: Representative coronal brain slice with fiber tip and GCaMP6s expression. White dashed line: BLA outline. Gray dashed rectangle: fiber tract. Blue: DAPI, Green: GCaMP6s. Scale 500 μm. Individual fiber placements are shown in *Figure 3—figure supplement 1A*. (**B**) Fluorescence traces from BLA of GCaMP6s-expressing CaMKIIα-Cre mice. On the last day PT (data shown for mouse 1), correct nose pokes (green line) were followed by a modest but significant rise in BLA CaMKIIα cell activity that increased steeply following receptacle entry (*Figure 3—figure supplement 1B*) while incorrect nose pokes (gray line) were followed by a modest decrease in activity. Mean Z%ΔF/F0 overlaid on bootstrapped 99% confidence intervals (99% bCIs). Shaded significance bars under traces represent time points where 99% bCIs do not contain 0 for at least 0.5 s. Correct: n = 44; downsampled incorrect: n = 44 of 141. Signal and reference channels (%ΔF/F0) during nose pokes are shown in *Figure 3—figure supplement 1C–D*. Incorrect nose pokes on the last day of PT versus Training Day 1 shown in *Figure 3—figure supplement 2B*. (**C**) Behavioral responses of CaMKIIα-Cre mice expressing GCaMP6s in BLA. Individual mice acquired the task at different rates as measured by rewards earned. Horizontal white line: acquisition threshold, when a mouse began to earn 20 rewards consistently in Training. Incorrect nose pokes shown in *Figure 3—figure supplement 2A*. (**D**) Heatmap of BLA CaMKIIα cell activity (Mouse 1) across all training phases, aligned to tone onset (Tone), correct nose poke (NP), and receptacle entry (Rec). Each row is the average of rewarded trials across a training session. White horizontal line: Day acquisition threshold met, as determined by rewards earned. Black horizontal lines: divisions between training phases. Black vertical lines: divisions between breaks in time to allow for variable latencies in tone onset, correct nose poke, and receptacle entry. Blanks in the heatmaps indicate time bins added for alignment. The bCI plot for Mouse 1 in *Figure 3—figure supplement 1G*. Individual heatmaps for mice 2–3 in *Figure 3—figure supplement 1E–F*. Incorrect nose pokes heatmaps for individual mice shown in *Figure 3—figure supplement 2C–E*. (**E**) Heatmap of BLA CaMKIIα cell activity averaged across mice. Signal aligned as in (D) with a selection of key days shown, from bottom to top: PT Day 1, PT Day 4, Early Training Day, Acquisition Day (white horizontal line), Last Extinction Day. The bCI plot for cohort averaged data in *Figure 3—figure supplement 1H*. Incorrect nose poke heatmaps averaged across mice shown in *Figure 3—figure supplement 2F*.

The online version of this article includes the following figure supplement(s) for figure 3:

**Figure supplement 1.** GCaMP6s recording in BLA CaMKIIα cells.

**Figure supplement 2.** GCaMP6s recording in BLA CaMKIIα cells: incorrect nose pokes.

**Figure supplement 3.** GCaMP6s recording in BLA CaMKIIα replicate.

**Figure supplement 4.** GCaMP6s recording in BLA CaMKIIα replicate: incorrect nose pokes.

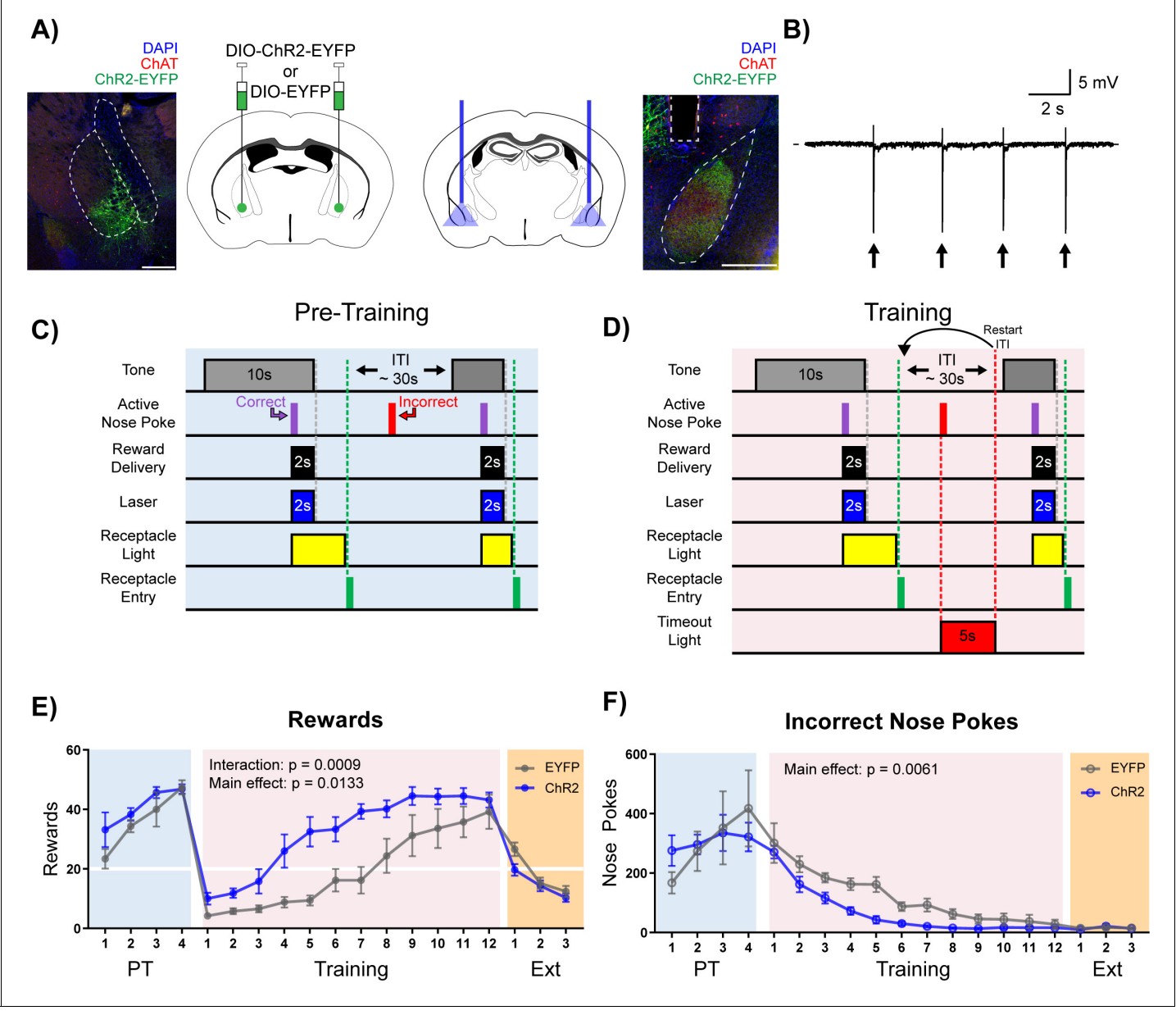

**Figure 4.** Stimulation of cholinergic terminal fibers in the BLA enhances cue-reward learning. (**A**) Schematic of optical stimulation of ChAT[+] terminal fibers projecting to the BLA. Left: Bilateral AAV injection into the NBM of ChAT-IRES-Cre mice to gain optical control over ChAT[+] NBM cells and representative coronal brain slice showing ChR2-EYFP expression. White dashed lines: internal capsule and globus pallidus outlines. Blue: DAPI, red: ChAT, green: ChR2-EYFP. Scale: 500 μm, individual injection sites shown in *Figure 4—figure supplement 1A* and separate channels shown in *Figure 4—figure supplement 1B*. Right: Bilateral optical fiber implantation above BLA to stimulate BLA-projecting ChAT[+] NBM cells. Representative coronal brain slice showing ChR2-EFYP expression and fiber tip placement. Gray dashed rectangle: fiber tract. White dashed: BLA outline. Blue: DAPI, red: ChAT, green: ChR2-EYFP. Scale: 500 μm, individual fiber tip placements shown in *Figure 4—figure supplement 1C* and separate channels shown in *Figure 4—figure supplement 1D*. Injection sites and fiber tip placements for males from *Figure 4—figure supplement 3C–F* shown in *Figure 4— figure supplement 4A–B*. (**B**) Optical stimulation validation *via* local field potential recordings. Extracellular recording of action potentials induced by optical stimulation of ChAT[+] NBM cells expressing ChR2. Arrows indicate 60 ms laser pulse. (**C–D**) Details of the Cue-Reward Learning Paradigm (**C**) During Pre-Training, auditory tones were presented on a variable interval 30 schedule (VI30), during which an active nose poke (correct) yielded Ensure reward delivery and 2 s of optical stimulation but there was no consequence for incorrect nose pokes (active nose pokes not during tone). (**D**) Training was identical to Pre-Training, except incorrect nose pokes resulted in a 5 s timeout, signaled by house light illumination, followed by a restarting of the ITI. (**E**) Behavioral performance in a cue-reward learning task improves with optical stimulation of ChAT[+] fibers in BLA. EYFP- and ChR2-expressing mice earn similar numbers of rewards during PT (blue shaded region). ChR2-expressing mice more rapidly earn significantly more rewards than EYFP-expressing mice during Training (pink shaded region). No significant differences were observed during extinction training (orange shaded region). Horizontal white line: acquisition threshold, when a mouse began to earn ~20 rewards consistently in Training. Mean ± SEM, EYFP: n = 5, ChR2: n = 6.
*Figure 4 continued on next page*

Figure 4 continued

Individual data are shown in *Figure 4—figure supplement 3A*. Data for males shown in *Figure 4—figure supplement 3C,E*. (F) EYFP- and ChR2-expressing mice made similar numbers of incorrect nose pokes during Pre-Training. ChR2-epxressing mice made significantly fewer incorrect nose pokes than EYFP-expressing mice in Training. No significant differences were observed during extinction training. Mean ± SEM, EYFP: n = 5, ChR2: n = 6. Individual data are shown in *Figure 4—figure supplement 3B*. Data for males shown in *Figure 4—figure supplement 3D,F*. Additional behavioral assays shown in *Figure 4—figure supplement 5A–F*.

The online version of this article includes the following figure supplement(s) for figure 4:

**Figure supplement 1.** Injection sites and optical fiber placements.
**Figure supplement 2.** Ex vivo electrophysiology.
**Figure supplement 3.** Individual behavioral data and male cohort.
**Figure supplement 4.** Injection sites and optical fiber placements.
**Figure supplement 5.** Additional behavioral assays with NBM-BLA optical stimulation.

rewarded trial: tone, correct nose poke, and reward retrieval, since these events were often separated by short latencies.

As seen in previous experiments, during the Pre-Training phase animals made a high number of nose poke responses over the course of each session, obtained most available rewards by the last day (*Figure 4E* + *Figure 4—figure supplement 3A*, blue shading), and committed a very high number of incorrect nose pokes (*Figure 4F* + *Figure 4—figure supplement 3B*, blue shading). There were no differences in rewards earned (main effect of group (EYFP versus ChR2) in a two-way repeated-measures ANOVA, $F_{(1, 9)}=1.733$, p=0.2205) or incorrect nose pokes (main effect of group (EYFP versus ChR2) in a two-way repeated-measures ANOVA, $F_{(1, 9)}=0.002433$, p=0.9617) between the EYFP control (n = 5) and ChR2 (n = 6) groups during the Pre-Training phase (*Figure 4E-F* + *Figure 4—figure supplement 3A–B*, blue shading), suggesting that increasing BLA ACh signaling was not sufficient to modify behavior during the Pre-Training phase of the task.

On Day 1 of the Training phase, all animals earned fewer rewards (*Figure 4E* + *Figure 4—figure supplement 3A*, pink shading) and incorrect nose pokes remained high (*Figure 4F* + *Figure 4—figure supplement 3B*, pink shading). As the animals learned that a nose poke occurring outside of the cued period resulted in a timeout, both control EYFP and ChR2 groups learned the contingency and improved their performance, resulting in acquisition of the cue-reward task (20 rewards earned). However, significant group differences emerged, such that ChR2 mice earned significantly more rewards than EYFP controls (*Figure 4E* + *Figure 4—figure supplement 3A*, pink shaded; main effect of group (EYFP versus ChR2) in a two-way repeated-measures ANOVA, $F_{(1, 9)}=9.434$, p=0.0133), and there was a significant Day × Group (EYFP versus ChR2) interaction (two-way repeated-measures ANOVA, $F_{(11, 99)}=3.210$, p=0.0009). ChR2 mice also made significantly fewer incorrect nose pokes than control mice (*Figure 4F* + *Figure 4—figure supplement 3B*, pink shaded; two-way repeated-measures ANOVA, $F_{(1, 9)}=12.67$, p=0.0061), suggesting that the ChR2 group learned the tone-reward contingency more quickly than the EYFP group. EYFP mice were able to reach the same peak cue-reward performance as the ChR2 group only after 4–6 additional days of training. Once peak performance was achieved, there was no difference in extinction learning between the groups (main effect of group (EYFP versus ChR2) in a two-way repeated-measures ANOVA, $F_{(1, 9)}=2.293$, p=0.1643). While sex differences in the behavior were not formally tested side by side, an independent cohort of male mice (EYFP n = 7, ChR2 n = 7, *Figure 4—figure supplement 4*) was tested to determine whether both male and female mice would respond to ACh stimulation, revealing similar trends during Training for rewards earned (*Figure 4—figure supplement 3C,E*, pink shaded; two-way repeated-measures ANOVA, Group main effect (EYFP versus ChR2): $F_{(1, 12)}=3.636$, p=0.0808, Day × Group interaction: $F_{(11, 132)}=3.033$, p=0.0012) and incorrect nose pokes (*Figure 4—figure supplement 3D,F*, red shaded; two-way repeated-measures ANOVA, Group main effect (EYFP versus ChR2): $F_{(1, 12)}=4.925$, p=0.0465).

In order to determine if optical stimulation of NBM-BLA cholinergic terminals improved performance in the task by increasing the rewarding value of the outcome, rather than enhancing cue-reward learning by some other means, we allowed mice to nose poke for optical stimulation rather than for Ensure (*Figure 4—figure supplement 5A*). There were no differences between the EYFP control and ChR2 groups (two-way repeated-measures ANOVA, $F_{(1, 9)}=0.6653$, p=0.4357). We also tested whether NBM-BLA cholinergic terminal activation was reinforcing on its own by stimulating

these terminals in a real-time place preference test. Mice were allowed to explore two similar compartments to determine baseline preference, and NBM-BLA cholinergic terminals were then stimulated in one of the two chambers to determine whether it increased time spent in the simulation-paired chamber. There was no difference between groups (*Figure 4—figure supplement 5B*, main effect of group (EYFP versus ChR2) in a two-way repeated-measures ANOVA, $F_{(1, 9)}=0.1311$, p=0.7257) in place preference, confirming that optical activation of NBM-BLA cholinergic terminals is not innately rewarding. Stimulation of NBM-BLA cholinergic terminals also did not lead to changes in nose poke behavior in an uncued progressive ratio task (*Figure 4—figure supplement 5C*, main effect of group (EYFP versus ChR2) in a two-way repeated-measures ANOVA, $F_{(1, 12)}=0.0009814$, p=0.975). Locomotor behavior was also not significantly affected by NBM-BLA cholinergic terminal activation (*Figure 4—figure supplement 5D*, two-way repeated-measures ANOVA, $F_{(1, 9)}=0.05804$, p=0.8150.) Finally, to determine whether there was any effect of NBM-BLA cholinergic terminal stimulation on preference for, or avoidance of, a stressful environment, mice were tested for changes in time spent in the dark or light side due to laser stimulation in the Light/Dark Box test, and there were no differences between the groups (*Figure 4—figure supplement 5E–F*, unpaired t-tests, number of crosses: p=0.3223; time in light side: p=0.1565).

## Muscarinic, but not nicotinic, receptors are required for acquisition of the cue-reward contingency

ACh signals through multiple receptor subtypes, with rapid, ionotropic signaling mediated through stimulation of nAChRs, and metabotropic signaling mediated through stimulation of mAChRs (*Picciotto et al., 2012*). To determine which ACh receptors were involved in this cue-reward learning task, mice were injected intraperitoneally with saline (n = 8), mecamylamine (non-competitive nicotinic antagonist, Mec, n = 9), scopolamine (competitive muscarinic antagonist, Scop, n = 8), or a combination of both antagonists (Mec+Scop, n = 9) 30 min prior to Pre-Training and Training, during the same epochs of the task in which optical stimulation was administered (*Figure 5A*). Like optical stimulation, blockade of ACh receptors during the Pre-Training phase of the task had no effect on rewards earned (*Figure 5B* + *Figure 5—figure supplement 1A*, blue shading, main effect of Group (antagonist) in a two-way repeated-measures ANOVA, $F_{(3, 30)}=1.285$, p=0.2973) or on the large number of incorrect nose pokes (*Figure 5C* + *Figure 5—figure supplement 1B*, blue shading, main effect of Group (antagonist) in a two-way repeated-measures ANOVA, $F_{(3, 30)}=1.496$, p=0.2356). By contrast, blockade of muscarinic signaling abolished the ability of mice to learn the correct cue-reward contingency during the Training period (*Figure 5B* + *Figure 5—figure supplement 1A*, pink shading, two-way repeated-measures ANOVA, Antagonist main effect: $F_{(3, 30)}=23.13$, p<0.0001, Day x Antagonist interaction: $F_{(33, 330)}=10.79$, p<0.0001), with these mice maintaining high levels of incorrect nose pokes for the duration of Training compared to Saline and Mec treated mice (*Figure 5C* + *Figure 5—figure supplement 1B*, pink shading, main effect of Group (antagonist) in a two-way repeated-measures ANOVA, $F_{(3, 30)}=25.64$, p<0.0001). Saline and Mec groups were not significantly different in any phase of the task, including across Extinction (*Figure 5B-C* + *Figure 5—figure supplement 1A–B*, orange shading, main effect of Group (antagonist) in a two-way repeated-measures ANOVA, $F_{(1, 15)}=1.201$, p=0.2903). We have shown that this dose of mecamylamine delivered i.p. has significant effects in tests of anxiety-like behavior and responses to inescapable stress. In addition, chronic treatment with mecamylamine at this dose is sufficient to decrease BLA c-fos immunoreactivity (*Mineur et al., 2007*). Consistent with the inability to acquire the cue-reward contingency, mice treated with Scop or Mec+Scop also obtained very few rewards during Extinction (*Figure 5B* + *Figure 5—figure supplement 1A*, orange shading). The antagonists had no effect on locomotion as measured by beam breaks (*Figure 5—figure supplement 1C*, one-way ANOVA, $F_{(3, 30)}=0.5074$, p=0.6802).

## ACh-mediated accelerated cue-reward learning does not require contingent stimulation of ChAT[+] NBM terminals in the BLA

Acetylcholine is often thought of as a neuromodulator (*Picciotto et al., 2012*), and the window for cholinergic effects on synaptic plasticity varies across ACh receptor subtypes (*Gu and Yakel, 2011*). It is therefore possible that ACh signaling may result in intracellular signaling changes that outlast the cue presentation window. In order to determine if the effect of NBM-BLA stimulation is

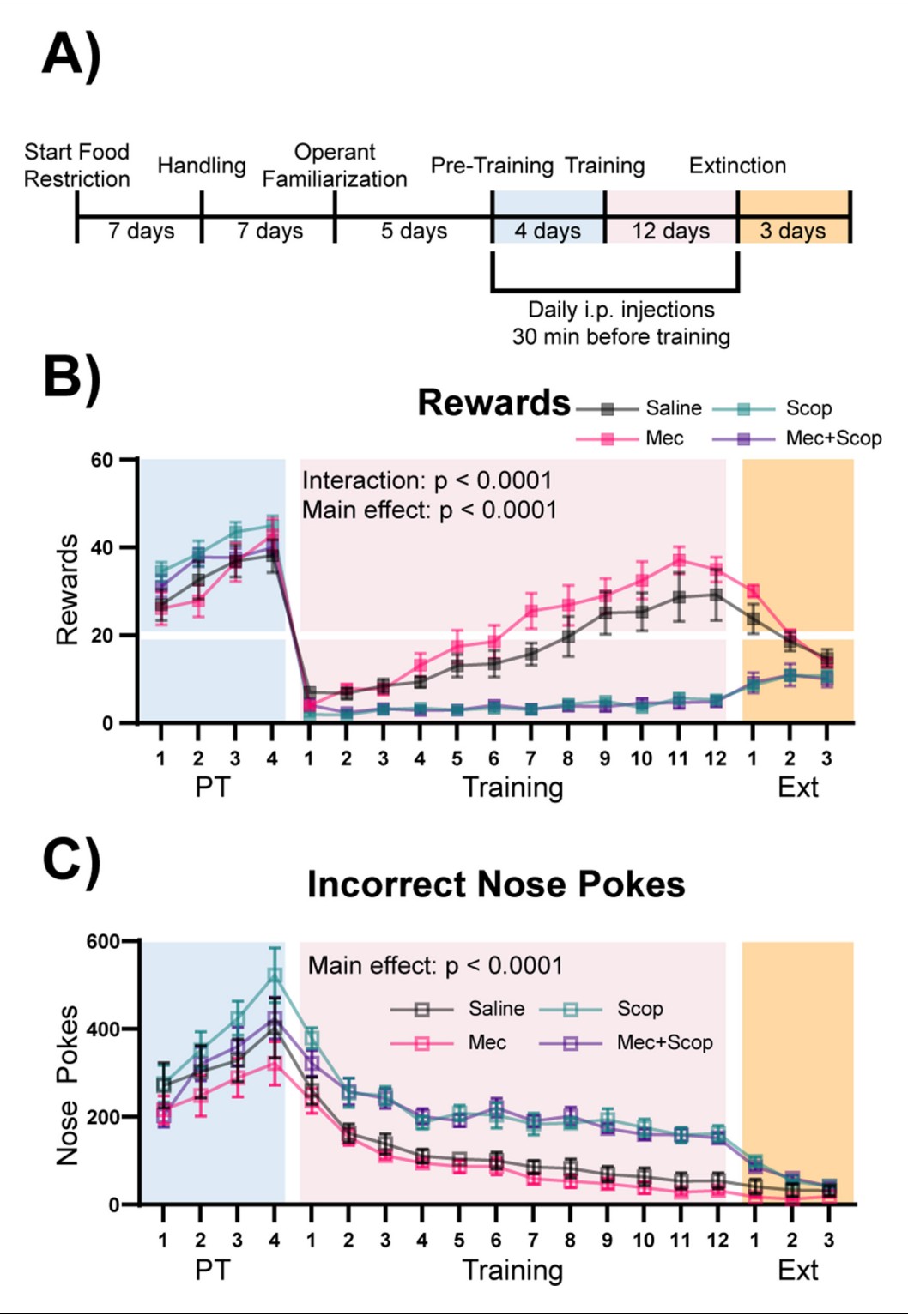

**Figure 5.** Muscarinic, but not nicotinic, ACh receptor antagonism prevents learning of a cue-reward contingency. (A) Timeline of drug administration. Saline or ACh receptor (AChR) antagonists were delivered i.p., 30 min before PT and Training sessions, the same phases of the task as optical stimulation in *Figure 4*. (B) Behavioral performance of mice administered AChR antagonists. AChR antagonists had no significant effect on rewards earned during Pre-Training. Muscarinic AChR antagonism (Scop and Mec+Scop) resulted in significantly fewer rewards earned during Training. There was no significant difference between saline controls and those receiving the nicotinic AChR antagonist (Mec) during Training and mice extinguished responding at similar rates.

*Figure 5 continued on next page*

*Figure 5 continued*

Mean ± SEM Saline (n = 8), Mec (n = 9), Scop (n = 8), Mec+Scop (n = 9). Horizontal white line: acquisition threshold, when a mouse began to earn ~20 rewards consistently in Training. Individual data are shown in *Figure 5—figure supplement 1A*. (C) Incorrect nose pokes. Incorrect nose poking was not affected by AChR antagonism during PT but Scop- and Scop+Mec-treated mice maintained high levels of incorrect nose pokes compared to Saline- and Mec-treated mice throughout Training. Mean ± SEM, Saline (n = 8), Mec (n = 9), Scop (n = 8), or Mec+Scop (n = 9). Individual data are shown in *Figure 5—figure supplement 1B*. AChR antagonist locomotor test shown in *Figure 5—figure supplement 1C*.

The online version of this article includes the following figure supplement(s) for figure 5:

**Figure supplement 1.** Individual behavioral data and locomotion.

dependent upon the timing of correct nose poke and laser stimulation contingency, we repeated the experiment in an independent cohort of mice with an additional non-contingent ChR2 group that received the same number of stimulation trains as the contingent ChR2 group, but in which light stimulation was explicitly unpaired with task events (*Figure 6A* + *Figure 6—figure supplement 1*). As in the previous experiment, there were no differences between the EYFP control (n = 6) and stimulation groups (contingent-ChR2 n = 5 and non-contingent ChR2 n = 5) during Pre-Training (*Figure 6B-C* + *Figure 6—figure supplement 2A–B*, blue shading; main effect of group (EYFP versus contingent-ChR2 versus non-contingent ChR2) two-way repeated-measures ANOVAs; rewards earned: $F_{(2, 13)}$=0.7008, p=0.5140; incorrect nose pokes: $F_{(2, 13)}$=0.3906, p=0.6843). However, the non-contingent ChR2 group was not significantly different from the contingent ChR2 group during the training period with respect to number of rewards earned (two-way repeated-measures ANOVA, $F_{(1, 8)}$=0.09147, p=0.7700) or incorrect nose pokes (two-way repeated-measures ANOVA, $F_{(1, 8)}$=0.3681, p=0.5609), but both ChR2 groups were significantly better than the EYFP control group (*Figure 6B-C* + *Figure 6—figure supplement 2A–B*, pink shading; two-way repeated-measures ANOVAs; rewards earned: Group (EYFP versus contingent-ChR2 versus non-contingent-ChR2) main effect: $F_{(2, 13)}$=7.254, p=0.0077; Day × Group interaction: $F_{(22, 143)}$=1.861, p=0.0164. Incorrect nose pokes: Group main effect: $F_{(2, 13)}$=4.884, p=0.0262). These results demonstrate that ChR2-mediated ACh release does not have to be time-locked to the cue, nose poke, or reward retrieval to improve performance of the task, suggesting that ACh may alter the threshold for neuronal plasticity for cue-reward pairing over a much longer timescale than might be expected based on results from the ACh3.0 recording and NBM-BLA recordings, which could be consistent with the involvement of mAChR signaling in this effect. As in the previous experiment, once all groups reached criterion for acquisition of the cue-reward contingency, there were no differences between any of the groups during Extinction (*Figure 6B-C* + *Figure 6—figure supplement 2A–B*, orange shaded; two-way repeated-measures ANOVA, $F_{(2, 13)}$=0.04229, p=0.9587).

## Discussion

It is increasingly recognized that the BLA is involved in learning to predict both positive and negative outcomes from previously neutral cues (*Cador et al., 1989*; *Janak and Tye, 2015*; *LeDoux et al., 1990*). Cholinergic cells in the basal forebrain complex fire in response to both positive and negative reinforcement (*Hangya et al., 2015*). The results shown here indicate that ACh signaling in the BLA is intimately involved in cue-reward learning. Endogenous ACh is released in the BLA in response to salient events in the task, and ACh dynamics evolved as the subject formed associations between stimuli and reward. While the pattern of ACh signaling in the BLA may seem reminiscent of how dopamine neurons encode reward prediction errors as measured in other brain areas (*Schultz et al., 1997*), the current results suggest that ACh release in the BLA may instead be involved in signaling a combination of salience and novelty. ACh release and NBM-BLA activity increased following correct nose poke and, around the time that animals acquired the cue-reward task, following tone onset. However, earlier in Training, incorrect nose pokes that resulted in a timeout were also followed by ACh release, although this was smaller in magnitude. Further, stimulating NBM-BLA cholinergic terminals during learning enhanced behavioral performance, but was not intrinsically rewarding on its own and did not support responding for the tone alone. Although ACh was released in the BLA at discrete points during the task, the effects of heightened BLA ACh signaling were relatively

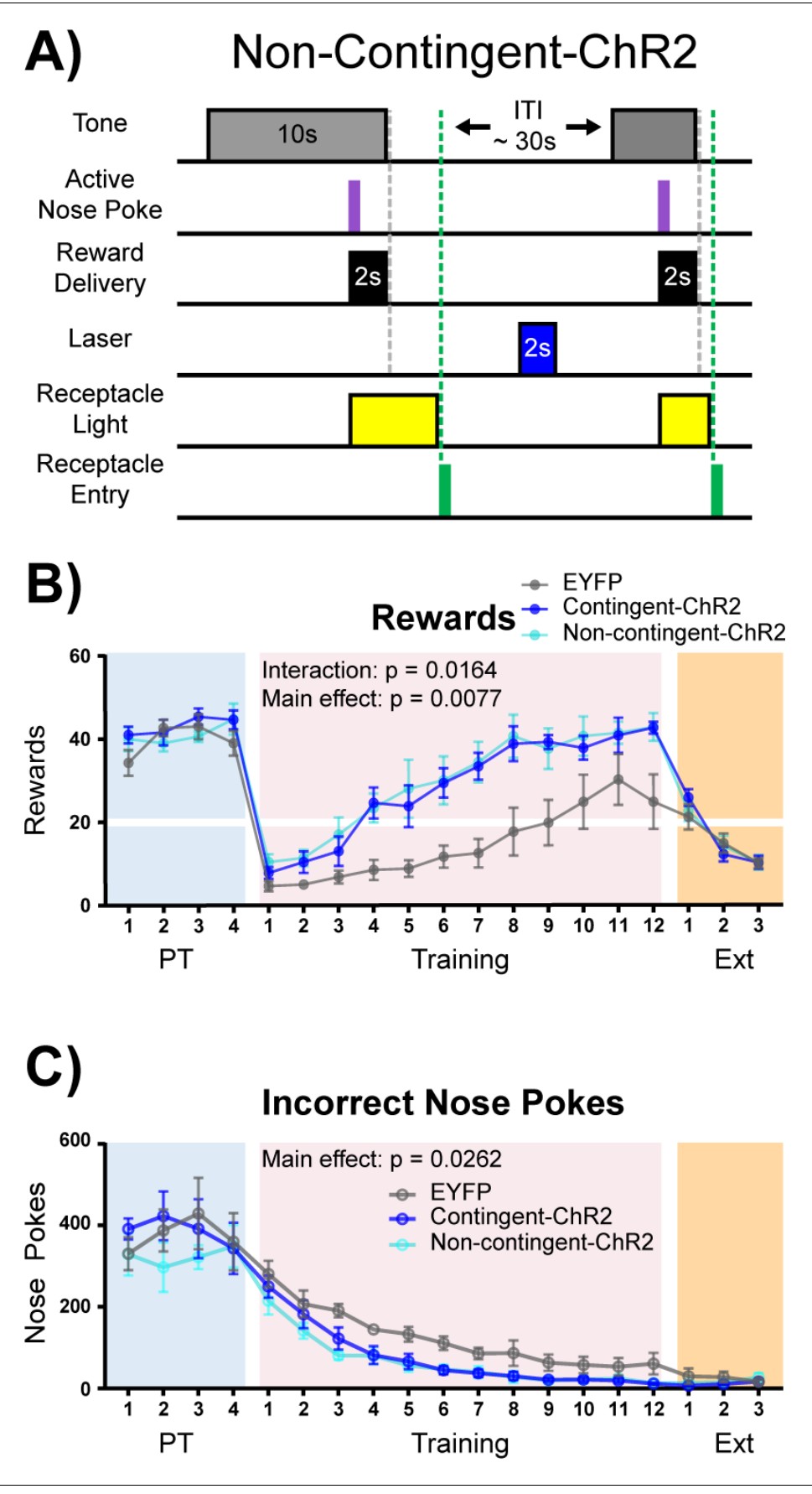

**Figure 6.** Non-contingent stimulation of cholinergic NBM-BLA terminals is sufficient to enhance cue-reward learning. (A) Experimental details of laser stimulation in non-contingent-ChR2 mice. Non-contingent-ChR2-expressing mice received the same number of light stimulations as contingent-ChR2-expressing mice, but stimulation was only given during the ITI, when non-contingent mice had not made a response within 2 s. Injection sites and fiber placements are shown in *Figure 6—figure supplement 1A–B*. (B) Non-contingent NBM-BLA optical stimulation also improves behavioral performance in cue-reward learning task. There was no significant difference in the number of rewards earned between EYFP (n = 6), contingent-ChR2 (n = 5), or non-contingent-ChR2 (n = 5) mice during Pre-Training. Contingent- and non-contingent-ChR2-expressing mice more rapidly earned significantly more rewards during Training than EYFP-expressing mice. No differences were observed between groups during extinction training. Mean ± SEM EYFP: n = 6, contingent-ChR2: n = 5, non-contingent-ChR2: n = 5. Horizontal white line: acquisition threshold, when a mouse began to earn 20 rewards consistently in Training. Individual data are shown in *Figure 6—figure supplement 2A*. (C) Incorrect nose pokes. There was no significant difference in the number of incorrect nose pokes between groups during Pre-Training. Contingent- and non-contingent-ChR2-expressing mice made significantly fewer incorrect nose pokes during Training than EYFP-expressing mice. No differences between groups were observed during extinction training. Mean ± SEM EYFP: n = 6, contingent-ChR2: n = 5, non-contingent: n = 5. Individual data are shown in *Figure 6—figure supplement 2B*.

The online version of this article includes the following figure supplement(s) for figure 6:

**Figure supplement 1.** Injection sites and optical fiber placements.
**Figure supplement 2.** Individual behavioral data.

long lasting, since it was not necessary for stimulation to be time-locked to cue presentation or reward retrieval to enhance behavioral performance. Thus, cholinergic inputs from the basal forebrain complex to the BLA are a key component of the circuitry that links salient events to previously neutral stimuli in the environment and uses those neutral cues to predict future rewarded outcomes.

## BLA ACh signaling and CaMKIIα cell activity are related to cue-reward learning

We have shown that ACh release in the BLA is coincident with the stimulus that was most salient to the animal at each phase of the task. Use of the fluorescent ACh sensor was essential in determining these dynamics (*Jing et al., 2018*; *Jing et al., 2019*). Previous microdialysis studies have shown that ACh is released in response to positive, negative, or surprising stimuli, but this technique is limited by relatively long timescales (min) and cannot be used to determine when cholinergic transients align to given events in an appetitive learning task and how they evolve over time (*Sarter and Lustig, 2020*). In this cue-reward learning paradigm, when there was no consequence for incorrect nose-poking (Pre-Training phase), animals learned to perform a very high number of nose pokes and received a large number of rewards, and BLA ACh signaling peaked following correct nose pokes. Both the behavioral response (nose poking that was not contingent with the tone) and the ACh response (linked to the correct nose poke) suggest that the animals were not attending to the tone during most of the Pre-Training phase of the task, but rather were attending to the cues associated with reward delivery, such as the reward light or the sound of the pump that delivered the reward. Consistent with this possibility, in the next phase of the task when mice received a timeout for responding if the tone was not presented, performance of all groups dropped dramatically. Interestingly, in animals that had difficulty learning the cue-reward contingency, during early Training sessions ACh release shifted to reward retrieval, likely because this was the most salient aspect of the task when the majority of nose pokes performed did not result in reward. Finally, as mice acquired the contingency between tone and reward availability, the tone also began to elicit ACh release in the BLA, suggesting that mice learned that the tone is a salient event predicting reward availability. Since there are multiple sources of ACh input to the BLA, it was important to determine whether NBM cholinergic neurons were active during the periods when ACh levels were high (*Woolf, 1991*). Recordings from cholinergic NBM-BLA terminal fibers showed similar dynamics to ACh measurements, suggesting that the NBM is a primary source of ACh across the phases of cue-reward learning.

Perhaps the most well-known example of dynamic responding related to learning cue-reward contingencies and encoding of reward prediction errors is the firing of dopaminergic neurons of the ventral tegmental area (VTA; *Schultz, 1998*). After sufficient pairings, dopaminergic neurons will fire

in response to the cue that predicts the reward, and no longer to the rewarding outcome, which corresponds with behavioral changes that indicate an association has been formed between conditioned stimuli (CS) and unconditioned stimuli (US). It should be noted that dopamine signaling is not unique in this learning-related dynamic response profile. Serotonergic neuronal responses also evolve during reward learning in a manner distinct from dopaminergic neurons (*Zhong et al., 2017*). Plasticity related to learning has also been observed in cholinergic neurons in the basal forebrain complex during aversive trace conditioning, such that after several training days, neuronal activity spans the delay between CS and US (*Guo et al., 2019*). Additionally, a recent study suggested that ACh may signal a valence-free reinforcement prediction error (*Sturgill et al., 2020*). Future studies on the selective inputs to NBM to BLA cholinergic neurons would be of interest to identify the links between brain areas involved in prediction error coding.

We found that BLA CaMKIIα cells were most reliably activated following reward retrieval before contingency acquisition (both when they were receiving several rewards but no timeouts in Pre-Training and few rewards early in Training). Similar to the recording of ACh levels, after acquisition, the tone began to elicit an increase in BLA CaMKIIα cell population activity. However, activity of CaMKIIα neurons differed from ACh signaling in the BLA in important ways. ACh was released in response to the salient events in the task that were best able to predict reward delivery or availability. By contrast, the activity of BLA CaMKIIα neurons was not tightly time-locked to correct nose poking, and instead followed reward retrieval until acquisition, when activity increased in response to tone onset. The divergent dynamics of ACh release and CaMKIIα neuron activity underscores that ACh's role in the BLA is to modulate, rather than drive, the activity of CaMKIIα neurons, and therefore may alter dynamics of the network through selective engagement of different populations of GABA interneurons (*Unal et al., 2015*).

## Increasing BLA acetylcholine levels enhances cue-reward learning

Neuronal activity and plasticity in the BLA is required for both acquisition of appetitive learning (conditioned reinforcement) and fear conditioning, however the inputs that increase activity in the structure during salient events likely come from many brain areas (*McKernan and Shinnick-Gallagher, 1997*; *Rogan et al., 1997*; *Tye et al., 2008*). In particular, dopaminergic inputs to the BLA are important for acquisition of conditioned reinforcement and for linking the rewarding properties of addictive drugs to cues that predict their availability (*Cador et al., 1989*). Our results indicate that ACh is a critical neuromodulator upstream of the BLA that is responsive to salient events, such as reward availability, motor actions that elicit reward, and cues that predict reward. We show here that increasing endogenous ACh signaling in the BLA caused mice to perform significantly better than controls in an appetitive cued-learning task. Heightened ACh release during learning of a cue-action-reward contingency led to fewer incorrect responses and increased acquisition rate in both female and male mice. The optical stimulation was triggered by correct nose poke, thus the cholinergic NBM-BLA terminal fiber stimulation overlapped with all three salient events: tone, nose poke, and reward retrieval, since the tone terminated 2 s after correct nose poke. We chose this stimulation pattern, as opposed to linking optical stimulation to tone onset, to ensure stimulation was dependent on behavioral responses. Therefore, stimulation did not precisely recapitulate the ACh release profile observed in mice that had already acquired the task (when ACh increases following tone onset). This suggests that behaviorally-contingent increases in BLA ACh are sufficient to enhance task acquisition (but see below). It is also possible that optogenetic-mediated ACh release may last longer than endogenous, tone-evoked release. A simultaneous stimulation and recording approach would be required to compare ACh release under both conditions (*Pisansky et al., 2019*). It is important to note that basal forebrain neurons have been demonstrated to co-release ACh and GABA (*Ma et al., 2018*; *Saunders et al., 2015*), and cholinergic basal forebrain neurons that project to the BLA have been shown to co-express a glutamate transporter (*Ma et al., 2018*; *Nickerson Poulin et al., 2006*). Thus, it is possible that both fiber photometry and optogenetic results could be influenced, in part, by co-release of other neurotransmitters from ChAT-positive neurons. Future studies employing additional fluorescent neurotransmitter sensors (*Marvin et al., 2013*; *Marvin et al., 2018*; *Marvin et al., 2019*) could help understand the interaction between the different signals employed by basal forebrain neurons.

It is possible that ACh improved learning by increasing the intensity of the reward, potentiating the learned association, improving discrimination, or a combination of these phenomena. However,

increasing ACh release in the BLA was not inherently rewarding, because it did not support self-stimulation or real-time place preference. This is at odds with a recent study that found stimulation of NBM-BLA cholinergic terminals could induce a type of place-preference and modest self-stimulation (*Aitta-aho et al., 2018*). Perhaps slight differences in targeting of ChR2 infusion or differences in the behavioral paradigm could be responsible for the lack of direct rewarding effects of optical ChAT terminal stimulation in the current study. Other recent work (*Jiang et al., 2016*) has demonstrated that stimulating this NBM-BLA cholinergic pathway is sufficient to strengthen cued aversive memory, suggesting that the effect of ACh in the BLA may not be inherently rewarding or punishing but instead potentiates plasticity in the BLA, allowing learning of cue-outcome contingencies. Similarly, it is possible that ACh alters motor activity. However, there were no effects of optical stimulation on locomotion or responding in the inactive nose poke port. In addition, during the Pre-Training phase when there was no consequence for incorrect nose pokes, all groups earned the same number of rewards, regardless of optical stimulation or pharmacological blockade of ACh receptors, suggesting that ACh is not involved in the motor aspects of the task or the value of the reward. Indeed, differences emerged only during the Training phase, when attention to the tone was critical to earn rewards. Further, incorrect nose poking remained high for mice administered scopolamine. This suggests that scopolamine-treated animals were seeking the reward, as in the operant familiarization and Pre-Training phases of training, but were unable to learn that they should only nose poke in response to the tone.

Cell-type-specific expression of AChRs and activity-dependent effects place cholinergic signaling at a prime position to shape BLA activity during learning. For instance, late-firing interneurons in the BLA exhibit nAChR-dependent EPSP's when no effect is seen on fast-spiking interneurons, while principal neurons can be either excited or inhibited through mAChRs, depending on activity level of the neuron at the time of cholinergic stimulation (*Unal et al., 2015*). BLA mAChRs can support persistent firing in principal neurons and can be important for the expression of conditioned place preference behavior, as well as trace fear conditioning (*Baysinger et al., 2012*; *Egorov et al., 2006*; *McIntyre et al., 1998*). Similar to studies of trace fear conditioning, in which activity of the network over a delay period must be maintained, we found that metabotropic (mAChRs) but not ionotropic (nAChRs) ACh receptors were required for learning the contingency of this cue-reward task. The timing of cholinergic signaling can be a critical factor in the induction of synaptic plasticity in other brain regions, so we hypothesized that the enhancement of cue-reward learning observed might be dependent upon when NBM-BLA terminal fibers were stimulated with respect to tone presentation and/or behavioral responses (*Gu and Yakel, 2011*). However, we found that heightened ACh signaling in the BLA improved behavioral performance even when stimulations were explicitly unpaired with the cue or correct nose poking. This suggests that the effect of increased cholinergic signaling in the BLA is long lasting, and that stimulation during a learning session is sufficient to potentiate synaptic events linking the cue to a salient outcome—even if CS and/or reward delivery are presented tens of seconds later. Given the findings from fiber photometry recordings, which showed endogenous ACh release was time-locked to salient stimuli during the task and evolved with learning, it is surprising that time-locking of exogenous ACh release was not necessary for enhancement of cue-reward learning. Coupled with pharmacological evidence demonstrating that muscarinic signaling is necessary for reward learning in this task, these results suggest the involvement of metabotropic signaling downstream of muscarinic receptors that outlasts the initial cholinergic stimulation.

To conclude, the abundant ACh input to the BLA results in ACh release in response to stimuli that predict reward in a learned cue-reward task. Increasing cholinergic signaling results in accelerated learning of the cue-reward contingency. These findings are consistent with the hypothesis that ACh is a neuromodulator that is released in response to salient stimuli and suggests that ACh signaling may enhance neuronal plasticity in the BLA network, leading to accelerated cue-reward learning.

## Materials and methods

**Key resources table**

| Reagent type (species) or resource | Designation | Source or reference | Identifiers | Additional information |
|---|---|---|---|---|

*Continued on next page*

*Continued*

| Reagent type (species) or resource | Designation | Source or reference | Identifiers | Additional information |
|---|---|---|---|---|
| Genetic reagent (*M. musculus*) | B6;129S6-Chat^tm2(cre)Lowl/J | Jackson Laboratory | Stock #: 006410 RRID:IMSR_JAX:006410 | |
| Genetic reagent (*M. musculus*) | C57BL/6J | Jackson Laboratory | Stock #: 000664 RRID:IMSR_JAX:000664 | |
| Genetic reagent (*M. musculus*) | Tg(*Camk2a-cre*)2Gsc | Günter Schütz, German Cancer Research Center | RRID:MGI:4457404 | *Casanova et al., 2001; Wohleb et al., 2016* |
| Antibody | Anti-ChAT (goat polyclonal) | Millipore Sigma | Cat #: AB144P RRID:AB_2079751 | (1:1000) |
| Antibody | Anti-GFP (chicken, polyclonal) | Thermo Fisher Scientific | Cat #: A10262, RRID:AB_2534023 | (1:1000) |
| Antibody | Anti-DsRed (rabbit, monoclonal) | Takara Bio | Cat #: 632392, RRID:AB_2801258 | (1:1000) |
| Antibody | Donkey anti-chicken 488 (secondary) | Jackson ImmunoResearch | Cat #: 703-545-155, RRID:AB_2340375 | (1:1000) |
| Antibody | Donkey anti-rabbit 555 (secondary) | Thermo Fisher Scientific | Cat #: A-31572, RRID:AB_162543 | (1:1000) |
| Antibody | Donkey anti-goat 555 (secondary) | Thermo Fisher Scientific | Cat #: A-21432, RRID:AB_141788 | (1:1000) |
| Antibody | Donkey anti-goat 647 (secondary) | Thermo Fisher Scientific | Cat #: A-21447, RRID:AB_141844 | (1:1000) |
| Recombinant DNA reagent | AAV9 hSyn-ACh3.0 | Yulong Li *Jing et al., 2018; Jing et al., 2019* | Cat #: YL10002-AV9 | |
| Recombinant DNA reagent | AAV1 Syn-FLEX-GCaMP6s-WPRE-SV40 | Addgene | Cat #: 100845-AAV1 RRID:Addgene_100845 | |
| Recombinant DNA reagent | AAV1-Syn-FLEX-jGCaMP7s-WPRE | Addgene | Cat #: 104491-AAV1 RRID:Addgene_104491 | |
| Recombinant DNA reagent | AAV1 Syn-FLEX-NES-jRCaMP1b-WPRE-SV40 | Addgene | Cat #: 100850-AAV1 RRID:Addgene_100850 | |
| Recombinant DNA reagent | AAV2 EF1a-DIO-EYFP | UNC Viral Vector Core | RRID:SCR_002448 | |
| Recombinant DNA reagent | AAV2 EF1a-DIO-hChR2(H134R)-EYFP | UNC Viral Vector Core | RRID:SCR_002448 | |
| Chemical compound, drug | Mecamylamine hydrochloride | Millipore Sigma | Cat #: M9020 | |
| Chemical compound, drug | (-) Scopolamine hydrochloride | Millipore Sigma | Cat #: S1013 | |
| Software, algorithm | MATLAB | MathWorks | RRID:SCR_001622 | Version 2020a |
| Software, algorithm | GraphPad Prism 8 | GraphPad Software | RRID:SCR_002798 | |
| Software, algorithm | EthoVision XT 10 | Noldus | RRID:SCR_000441 | |
| Software, algorithm | FV10-ASW | Olympus | RRID:SCR_014215 | Version 04.02.03.06 |
| Software, algorithm | Doric Neuroscience Studio | Doric Lenses | | Version 5.3.3.14 |
| Software, algorithm | MED-PC IV | Med Associates Inc | RRID:SCR_012156 | |
| Other | Allen Reference Atlas | *Lein et al., 2007* | RRID:SCR_013286 | |
| Other | DAPI stain | Thermo Fisher Scientific | Cat #: 62248 | 1:1000 |

## Animals

All procedures were approved by the Yale University Institutional Animal Care and Use Committee (protocol: 2019–07895) in compliance with the National Institute of Health's Guide for the Care and

Use of Laboratory Animals. Experiments were performed in mice of both sexes, in keeping with the NIH policy of including sex as a biological variable. Sex of mice in behavioral graphs is indicated by circles for females and squares for males.

Female and male heterozygous mice with Cre recombinase knocked into the choline acetyltransferase (ChAT) gene (ChAT-IRES-Cre, B6;129S6-Chat$^{tm2(cre)Lowl/J}$, Jackson Laboratory, Bar Harbor, ME) were bred in house by mating ChAT-IRES-Cre with C57BL6/J mice. CaMKIIα-Cre (Tg(Camk2a-cre)2Gsc) mice obtained from Ronald Duman (*Casanova et al., 2001*; *Wohleb et al., 2016*) were bred in house as above. C57BL6/J mice were obtained from The Jackson Laboratory at 6–10 weeks of age, and tested at 5–7 months of age, following at least 1 week of acclimation. All mice were maintained in a temperature-controlled animal facility on a 12 hr light/dark cycle (lights on at 7:00 AM). Mice were group housed 3–5 per cage and provided with ad libitum food and water until undergoing behavioral testing. Mice were single housed 1–3 weeks before surgery to facilitate food restriction and body weight maintenance.

## Surgical procedures

Surgical procedures for behavior were performed in fully adult mice at 4–6 months of age, age-matched across conditions. For viral infusion and fiber implantation, mice were anesthetized using isoflurane (induced at 4%, maintained at 1.5–2%) and secured in a stereotactic apparatus (David Kopf Instruments, Tujunga, CA). The skull was exposed using a scalpel and Bregma was determined using the syringe needle tip (2 µL Hamilton Neuros syringe, 30 gauge needle, flat tip; Reno, NV).

For fiber photometry surgeries, 0.4 µL of AAV9 hSyn-ACh3.0 (Vigene Biosciences Inc) to measure BLA ACh levels (*Figure 2A-E* + *Figure 2—figure supplements 1–2*) was delivered unilaterally to the BLA (A/P; −1.34 mm, M/L + or - 2.65 mm, D/V −4.6 mm, relative to Bregma) of ChAT-IRES-Cre or wild-type C57BL6/J mice at a rate of 0.1 µL/min. The needle was allowed to remain at the infusion site for 5 min before and 5 min after injection. A mono fiber-optic cannula (1.25 mm outer diameter metal ferrule; 6 mm long, 400 µm core diameter/430 µm outer diameter, 0.48 numerical aperture (NA), hard polymer cladding outer layer cannula; Doric Lenses, Quebec City, Quebec, Canada) was implanted above the BLA (A/P; −1.34 mm, M/L + 2.65 mm, D/V −4.25 mm) and affixed to the skull using opaque dental cement (Parkell Inc, Edgewood, NY). For BLA CaMKIIα cell calcium dynamic recordings (*Figure 3* + *Figure 3—figure supplements 1–2*), 0.5 µL of AAV1 Syn-FLEX-GCaMP6s-WPRE-SV40 (Addgene, Watertown, MA) was injected into the left BLA using the same procedure and coordinates but was injected into CaMKIIα-Cre mice. Cholinergic NBM-BLA terminal fiber calcium dynamic recording (*Figure 2F-J* + *Figure 2—figure supplements 5–8*) surgeries were performed as above except AAV1-Syn-FLEX-jGCaMP7s-WPRE (Addgene) was infused unilaterally into the NBM (A/P: - 0.7 mm, M/L + or - 1.75 mm, D/V – 4.5 mm) of ChAT-IRES-Cre mice, with the optical fiber being placed above the ipsilateral BLA. The jRCaMP1b + ACh3.0 surgeries to simultaneously measure cholinergic NBM-BLA terminal fiber calcium dynamics and BLA ACh levels (*Figure 2—figure supplement 9*) consisted of both the NBM and BLA infusions above, except AAV1 Syn-FLEX-NES-jRCaMP1b-WPRE-SV40 (Addgene) was infused the NBM of ChAT-IRES-Cre mice. The RCaMP sham mouse (*Figure 2—figure supplement 9E,H*) was a wild-type littermate and thus had no jRCaMP1b expression. pAAV.Syn.Flex.GCaMP6s.WPRE.SV40 (Addgene viral prep # 100845-AAV1; http://n2t.net/addgene:100845; RRID:Addgene_100845), pGP-AAV-syn-FLEX-jGCaMP7s-WPRE was a gift from Douglas Kim and GENIE Project (Addgene viral prep # 104491-AAV1; http://n2t.net/addgene:104491; RRID:Addgene_104491), and pAAV.Syn.Flex.NES-jRCaMP1b.WPRE.SV40 (Addgene viral prep # 100850-AAV1; http://n2t.net/addgene:100850; RRID:Addgene_100850) were gifts from Douglas Kim and GENIE Project (*Chen et al., 2013*; *Dana et al., 2016*; *Dana et al., 2019*).

Mice were allowed to recover in a cage without bedding with a microwavable heating pad underneath it until recovery before being returned to home cage. For 2 d following surgery, mice received 5 mg/Kg Rimadyl i.p (Zoetis Inc, Kalamazoo, MI) as postoperative care.

For optical stimulation experiments (*Figure 4*, *Figure 6* + *Figure 4—figure supplements 1–5* + *Figure 6—figure supplements 1–2*), surgeries were performed as above except as follows: 0.5 µL of control vector (AAV2 EF1a-DIO-EYFP) or channelrhodopsin (AAV2 EF1a-DIO-hChR2(H134R)-EYFP; University of North Carolina Gene Therapy Center Vector Core, Chapel Hill, NC) was delivered bilaterally into the NBM (A/P: - 0.7 mm, M/L ± 1.75 mm, D/V – 4.5 mm) of ChAT-IRES-Cre mice. Mono fiber-optic cannulas (1.25 mm outer diameter zirconia ferrule; 5 mm long, 200 µm core

diameter/245 µm outer diameter, 0.37 NA, polyimide buffer outer layer cannula; Doric Lenses) were inserted bilaterally above the basolateral amygdala (BLA, A/P; −1.22 mm, M/L ± 2.75 mm, D/V −4.25 mm). Mice were randomly assigned to EYFP or ChR2 groups, controlling for average group age.

For ex vivo local field potential electrophysiology experiments (*Figure 4B*), the NBM was injected with DIO-ChR2-EYFP as described above, except mice were 8 weeks of age (see Supplemental Methods for current clamp recording methods). The coronal brain slices containing the NBM were prepared after 2–4 weeks of expression. Briefly, mice were anesthetized with 1× Fatal-Plus (Vortech Pharmaceuticals, Dearborn, MI) and were perfused through their circulatory systems to cool down the brain with an ice-cold (4°C) and oxygenated cutting solution containing (mM): sucrose 220, KCl 2.5, $NaH_2PO_4$ 1.23, $NaHCO_3$ 26, $CaCl_2$ 1, $MgCl_2$ 6 and glucose 10 (pH 7.3 with NaOH). Mice were then decapitated with a guillotine immediately; the brain was removed and immersed in the ice-cold (4°C) and oxygenated cutting solution to trim to a small tissue block containing the NBM. Coronal slices (300 µm thick) were prepared with a Leica vibratome (Leica Biosystems Inc, Buffalo Grove, IL) after the tissue block was glued on the vibratome stage with Loctite 404 instant adhesive (Henkel Adhesive Technologies, Düsseldorf, Germany). After preparation, slices were maintained at room temperature (23–25°C) in the storage chamber in the artificial cerebrospinal fluid (ACSF; bubbled with 5% $CO_2$% and 95% $O_2$) containing (in mM): NaCl 124, KCl 3, $CaCl_2$ 2, $MgCl_2$ 2, $NaH_2PO_4$ 1.23, $NaHCO_3$ 26, glucose 10 (pH 7.4 with NaOH) for recovery and storage. Slices were transferred to the recording chamber and constantly perfused with ACSF with a perfusion rate of 2 mL/min at a temperature of 33°C for electrophysiological experiments. Cell-attached extracellular recording of action potentials was performed by attaching a glass micropipette filled with ACSF on EYFP-expressing cholinergic neurons with an input resistance of 10–20 MΩ under current clamp. Blue light (488 nm) pulse (60 ms) was applied to the recorded cells through an Olympus BX51WI microscope (Olympus, Waltham, MA) under the control of the Sutter filter wheel shutter controller (Lambda 10–2, Sutter Instrument, Novato, CA). All data were sampled at 3–10 kHz, filtered at 3 kHz and analyzed with an Apple Macintosh computer using Axograph X (AxoGraph). Events of field action potentials were detected and analyzed with an algorithm in Axograph X as reported previously (*Rao et al., 2008*).

## Behavioral testing

### Habituation

One week after surgery, mice were weighed daily and given sufficient food (2018S standard chow, Envigo, Madison, WI) to maintain 85% free-feeding body weight. All behavioral tests were performed during the light cycle. Mice were allowed to acclimate to the behavioral room for 30 min before testing and were returned to the animal colony after behavioral sessions ended.

Two weeks after surgery, mice were handled 3 min per day for 7 d in the behavioral room. Mice were given free access to the reward (EnsurePlus Vanilla Nutrition Shake solution mixed with equal parts water (Ensure); Abbott Laboratories, Abbott Park, IL) in a 50 mL conical tube cap in their home cages on the last 3 d of handling to familiarize them to the novel solution. Mice were also habituated to patch cord attachment during the last 3 d of handling for optical stimulation and fiber photometry experiments. Immediately before training each day, a patch cord was connected to their optical fiber(s) via zirconia sleeve(s) (1.25 mm, Doric Lenses) before being placed in the behavioral chamber.

### Operant training

All operant training was carried out using Med Associates modular test chambers and accessories (ENV-307A; Med Associates Inc, Georgia, VT). For optical stimulation experiments, test chambers were housed in sound attenuating chambers (ENV-022M). Two nose poke ports (ENV-313-M) were placed on the left wall of the chamber and the reward receptacle (ENV-303LPHD-RL3) was placed on the right wall. The receptacle cup spout was connected to a 5 mL syringe filled with Ensure loaded in a single speed syringe pump (PHM-100). Nose pokes and receptacle entries were detected by infrared beam breaks. The tone generator (ENV-230) and speaker (ENV-224BM) were placed outside the test chamber, but within the sound attenuating chamber, to the left. The house light (used for timeout, ENV-315M) was placed on top of the tone generator to avoid snagging patch cords. Each chamber had a fan (ENV-025F28) running throughout the session for ventilation and white noise. Behavior chambers were connected to a computer running MEDPC IV to collect

event frequency and timestamps. For optical stimulation experiments, a hole drilled in the top of the sound attenuating chambers allowed the patch cord to pass through. Initial BLA ACh3.0 (*Figure 2A–E*) and BLA CaMKIIα GCaMP6s (*Figure 3*) fiber photometry recordings occurred in a darkened behavioral room outside of sound attenuating chambers due to steric constraints with rigid fiber photometry patch cords. Later behavioral chamber optimization (wall height was extended with 3D printed and laser cut acrylic panels to allow removing the test chamber lid while preventing escape) allowed all other fiber photometry cohorts to be tested inside sound attenuating chambers. For fiber photometry experiments, a custom receptacle was 3D printed that extended the cup beyond the chamber wall to allow mice to retrieve the reward with more rigid patch cords. Each mouse was pseudo-randomly assigned to behavioral chamber when multiple chambers were used, counterbalancing for groups across boxes.

Three weeks after surgery, initial operant familiarization consisted of one 35 min session of Free Reward to demonstrate the location of reward delivery; all other sessions were 30 min. During Free Reward, only the reward receptacle was accessible. After 5 min of habituation, Ensure (24 μL over 2 s) was delivered in the receptacle cup and a light was turned on above the receptacle. The receptacle light was turned off upon receptacle entry. The next phase of operant familiarization, mice learned to nose poke to receive reward on a fixed-ratio one (FR1) schedule of reinforcement. Mice in experiments involving manipulations (optical stimulation and antagonist studies) were pseudo-randomly assigned to left or right active (reinforced) nose poke port. Mice in fiber photometry experiments were all assigned to right active port to minimize potential across subject variability. The inactive (unreinforced) port served as a locomotor control. During FR1 operant familiarization, each nose poke response into the active port resulted in receptacle light and reward delivery. After the mice reached criterion on FR1 operant familiarization (group average of 30 rewards for 2 consecutive days, usually 4–5 d), mice were advanced to the Pre-Training phase. This phase incorporated an auditory tone (2.5–5 kHz, ~60 dB) that lasted for at most 10 s and signaled when active nose pokes would be rewarded. Only active nose pokes made during the 10 s auditory tone (correct nose pokes) resulted in reward and receptacle light delivery. The tone co-terminated with Ensure delivery. During Pre-Training, there was no consequence for improper nose pokes, neither in the active port outside the tone (incorrect nose pokes) nor in the inactive port (inactive nose pokes). The number of inactive nose pokes were typically very low after operant familiarization and were not included in analysis. After reward retrieval (receptacle entry following reward delivery) the receptacle light was turned off and the tone was presented again on a variable intertrial interval schedule with an average interval of 30 s (VI 30), ranging from 10 to 50 s (*Ambroggi et al., 2008*). After 4–5 d of Pre-Training, mice progressed to the Training phase, which had the same contingency as Pre-Training except incorrect nose pokes resulted in a 5 s timeout signaled by house light illumination, followed by a restarting of the previous intertrial interval. Mice were considered to have acquired the task after earning 20 rewards during the Training phase of the task. In order to promote task acquisition, mice that were not increasing number of rewards earned reliably were moved to a VI 20 schedule after 9 d of VI 30 Training for BLA ACh3.0 or 6–7 d for BLA CaMKIIα cell mice. The VI 20 schedule was only needed for the two groups that were trained outside of the sound attenuating chambers. Mice progressed to Extinction after 12 d of Training or, in the case of fiber photometry cohorts, once all mice met the acquisition criteria. Extinction was identical to Training except no Ensure was delivered in response to correct nose pokes. The replicate cohorts of the BLA CaMKIIα GCaMP6s and NBM-BLA terminal fiber recording experiments were advanced to 1 d of Extinction after only 7 d of Training due to the COVID-19 shutdown.

Between mice, excrement was removed from the chambers with a paper towel. At the end of the day chambers were cleaned with Rescue Disinfectant (Virox Animal Health, Oakville, Ontario, Canada) and Ensure syringe lines were flushed with water then air. Mice were excluded from analyses if a behavioral chamber malfunctioned (e.g. syringe pump failed) or they received the improper compound. Fiber photometry mice were excluded from analyses if they did not meet the acquisition criterion by the last day of Training. See *Supplementary file 1* for number of mice that acquired, were excluded, and further explanations for behavioral paradigm deviations.

## Optical stimulation

Optical stimulation was generated by a 473 nm diode-pumped solid-state continuous wave laser (Opto Engine LLC, Midvale, UT) controlled by a TTL adapter (SG-231, Med Associates Inc). The laser was connected to a fiber optic rotary joint (Doric Lenses) via a mono fiber optic patch cord (200 μm core, 220 μm cladding, 0.53 NA, FC connectors; Doric Lenses). The rotary joint was suspended above the sound attenuating chamber with a connected branching fiber optic patch cord (200 μm core, 220 μm cladding, 0.53 NA, FC connector with metal ferrule; Doric Lenses) fed into the behavioral box. Laser power was adjusted to yield 10–12 mW of power at each fiber tip. The stimulation pattern was 25 ms pulses at 20 Hz for 2 s modified from parameters in *Jiang et al., 2016*. Jiang et al. used a 20 Hz pulse frequency, 5 ms pulses, and 10–12 mW power at the fiber tips. In this study, we used a 2 s stimulation duration because it matched the time of syringe pump activation for reward delivery and co-terminated with the pump and auditory tone. A 25 ms pulse width was used because our lasers were not able to generate sufficient power with 5 ms pulses. Optical stimulation was only delivered during the Pre-Training and Training phases of the operant task. Both control (EYFP) and experimental (ChR2) groups received identical light delivery, and stimulation was triggered by a correct nose poke and co-terminated with the auditory tone and Ensure delivery. For the non-contingent experiment, the number of light stimulations was yoked to the concurrently running ChR2 mouse. The timing of the non-contingent stimulation was explicitly unpaired with correct nose pokes or tones and was held in queue until the mouse had not made a response in the last 2 s, a tone was not going to be delivered within the next 2 s, or at least 5 s had passed since the mouse entered the receptacle after earning reward.

## Fiber photometry

### Acquisition

Fluorescent measurements of ACh and calcium levels were recorded using two Doric Lenses 1-site Fiber Photometry Systems: a standard 405/465 nm system and a 405/470/560 nm system. The standard 405/465 system was configured as follows: the fiber photometry console controlled the two connectorized LEDs (CLEDs, 405 nm modulated at 208.616 Hz and 465 nm modulated at 572.205 Hz) through the LED module driver (*Cassidy et al., 2019*). Each CLED was connected via attenuating patch cord to the five-port Fluorescence MiniCube (FMC5_AE(405)_AF(420-450)_E1(460–490)_F1 (500–550)_S). A pigtailed fiber optic rotary joint was connected to the MiniCube and suspended above the behavioral chamber with a rotary joint holder in order to deliver and receive light through the implanted optical fiber. The other end of the rotary joint was connected to the mono fiber optic patch cord via M3 connector and attached with a zirconia sleeve to the implanted fiber optic as above. The F1 (500–550 nm) port of the MiniCube was connected to the photoreceiver (AC low mode, New Focus 2151 Visible Femtowatt Photoreceiver, New Focus, San Jose, CA) via a fiber optic adapter (Doric Lenses) that was finally connected back to the fiber photometry console through an analog port. The 405/470/560 nm system was set up similarly, except a 560 nm LED was incorporated (modulated at 333.786 Hz), a six-port MiniCube with two integrated photodetector heads was used (iFMC6_IE(400-410)_E1(460–490)_F1(500–540)_E2(555–570)_F2(580–680)_S), and Doric Fluorescence Detector Amplifiers were used (AC 1× or 10× mode, DFD_FOA_FC). A TTL adapter (SG-231, Med Associates Inc) was connected to the digital input/output port to allow for timestamping when events occurred in the behavioral chamber. Signal was recorded using Doric Neuroscience Studio (V 5.3.3.14) via the Lock-In demodulation mode with a sampling rate of 12.0 kS/s. Data were decimated by a factor of 100 and saved as a comma-separated file.

## Pre-processing

Pre-processing of raw data was performed using a modified version of a MATLAB (MathWorks, Natick, MA) script provided by Doric. The baseline fluorescence (F0) was calculated using a first-order least mean squares regression over the ~30 min recording session. Second-order least mean squares regressions were used when photobleaching of the sensor was more pronounced, as in the case of NBM-BLA terminal fiber recordings. The change in fluorescence for a given time point (ΔF) was calculated as the difference between it and F0, divided by F0, which was multiplied by 100 to yield %ΔF/F0. The %ΔF/F0 was calculated independently for both the signal (465 nm) and reference (405 nm) channels to assess the degree of movement artifact. Since little movement artifact was

observed in the recordings (*Figure 2—figure supplement 1B–C*, *Figure 2—figure supplement 5D–E*, *Figure 3—figure supplement 1C–D*, tan lines), the signal %ΔF/F0 was analyzed alone (the code provided allows for running the entire analysis pipeline with the reference channel %ΔF/F0 if desired). The %ΔF/F0 was z-scored to give the final Z%ΔF/F0 reported here. For the BLA CaMKIIα cell recordings (*Figure 3—figure supplement 1C–D*), the reference channel displayed some mirroring (moving in the opposite direction) compared to the signal. This is likely because 405 nm is not the 'true' isosbestic point for GCaMP and we were instead measuring some changes in calcium-unbound GCaMP rather than calcium-insensitive GCaMP signal alone (*Barnett et al., 2017*; *Kim et al., 2016a*; *Sych et al., 2019*). Graphs and heatmaps for averaged traces aligned to actions were based on licking bout epoch filtering code from TDT (Alachua, FL; link in code comments).

## Heatmaps

Combined action heatmaps were generated in MATLAB (2020a) by analyzing data 5 s preceding tone onset (rewarded trials only) to 5 s after receptacle entry. Actions were aligned despite variable latencies by evenly splitting a maximum of 4 s post-tone onset/pre-correct nose poke and 1 s post-correct nose poke/pre-receptacle entry for each trial within a day. The resulting aligned trials were averaged to generate daily averages that made up the rows of the individual animal heatmaps. Blanks in the rows of heatmaps (black time bins) indicate time bins added for alignment, meaning that no trials for that day had a latency that stretched the entire window. Only rewarded trials where the mouse entered the receptacle within 5 s after nose poke were analyzed. Full or partial training days were excluded from analysis if there were acquisition issues such as the patch cord losing contact with the fiber or behavioral apparatus malfunction. Lack of trials (fewer than three) for analysis or recording issues led to missing rows of fiber photometry data in the heatmap despite having behavioral data, in which case these rows were skipped rather than adding entire blank rows. Due to individual differences in behavior, across-mouse average data was calculated by using a selection of days in which behavior was roughly similar or milestones such as the first and last day of Pre-Training, first day earning 10 rewards in Training, first day crossing acquisition threshold (and maintaining afterward), last day of Training, last day of Extinction (with three or more rewarded trials that met analysis criteria). Additional days were included in across-mouse average heatmaps when possible. Incorrect nose poke heatmaps were generated by averaging signals for 5 s before and 5 s after incorrect nose pokes that were not preceded by an incorrect nose poke in the last 5 s. The incorrect nose poke heatmaps averaged across mice were generated using the same selection of days as the combined action heatmaps for a given experiment.

## Bootstrapped confidence interval analyses

Bootstrapped confidence intervals (bCIs) of the Z-scored % ΔF/F0 fiber photometry data within and across mice were generated using the methods described in *Jean-Richard-Dit-Bressel et al., 2020* for the following events: tone onset, correct nose poke, receptacle entry, and incorrect nose poke. For the within-mouse analysis by day, trials were aligned to event onset, and bCIs were generated for events that had at least 3 trials from 5 s before to 10 s after each event. Each series of data were bootstrapped 10,000 times and a two-sided 99% confidence interval was constructed. Data were considered significantly different from baseline (Z% ΔF/F0 = 0) when their 99% bCIs did not contain zero for an interval of time designated by a consecutive threshold of 0.5 s.

To avoid comparing vastly different numbers of trials, in graphs where correct and incorrect nose pokes were plotted together, incorrect nose pokes were downsampled to match the number of correct nose poke trials. For Incorrect Nose Pokes graphs where last Pre-Training Day and Training Day 1 were plotted together, both days were downsampled to the number of correct nose pokes on the last Pre-Training Day.

For the combined action bCI plots (tone onset, correct nose poke, and receptacle entry), the selection of days for each mouse matched that of the cohort-averaged combined action heatmaps. The three-event plots were combined by cropping to match the maximum latencies used in the combined action heatmaps. For the across-mouse averaged bCI plots, analyses were carried out as above except the bootstrapping used mouse trial averages. The mean lines for across-mouse averaged bCI plots were calculated by taking the mean of all individual trials together. The NBM-BLA cholinergic terminal fiber experiment required combining the two independent cohorts to obtain

n $\geq$ 3. For the incorrect nose poke bCI plots, the number of trials used for each day was downsampled to 20 if a mouse performed more than 20.

## Pharmacology

Male wild-type C57BL/6J mice were injected i.p. 30 min before each Pre-Training and Training session with a volume of 10 mL/kg with the following compounds: 1× DPBS (Thermo Fisher Scientific, Waltham, MA), 1 mg/kg mecamylamine hydrochloride (Millipore Sigma, St. Louis, MO), 0.5 mg/kg (-) scopolamine hydrochloride (Millipore Sigma), or 1 mg/kg mecamylamine + 0.5 mg/kg scopolamine (*Figure 5* + *Figure 5—figure supplement 1*).

## Histology

After completion of behavioral experiments, animals were anesthetized with 1× Fatal-Plus (Vortech Pharmaceuticals). Once there was no response to toe-pinch, mice were transcardially perfused with 20 mL ice-cold 1× DPBS followed by 20 mL 4% paraformaldehyde (PFA, Electron Microscopy Sciences, Hatfield, PA). Brains were extracted and post-fixed for at least 1 d in 4% PFA at 4°C and transferred to 30% sucrose (Millipore Sigma) for at least 1 dy at 4°C. Brains were sliced 40 μm thick on a self-cooling microtome and stored in a 0.02% sodium azide (Millipore Sigma) PBS solution. Brain slices were washed in PBS, blocked for 2–3 hr (0.3% Triton X-100, American Bioanalytical, Canton, MA; 3% normal donkey serum, Jackson ImmunoResearch, West Grove, PA), then incubated overnight with primary antibodies (1:1000 + 1% normal donkey serum). Slices were then washed in PBS and incubated with secondary antibodies (1:1000) for 2 hr, washed, stained with DAPI for 5 min, washed, mounted, and coverslipped with Fluoromount-G (Electron Microscopy Sciences). All incubations were at room temperature. Microscope slides were imaged using a FLUOVIEW FV10i confocal microscope (Olympus). Injection sites and fiber placements were designated on modified Allen Mouse Brain Atlas figures (*Lein et al., 2007*). Mice were excluded from analyses if fluorescence was not observed at injection sites or if fiber tips were not identified at the intended site.

## Statistical analyses of behavior

Operant behavioral data saved by MEDPC IV was transferred to Excel using MPC2XL. Data were organized in MATLAB and analyzed in Prism (V8.3.0, GraphPad Software, San Diego, CA). Differences between groups and interactions across days for Training were evaluated using Two-Way Repeated Measures ANOVAs. We computed the required sample size for a 90% power level with an alpha of 0.05 by estimating the control (EYFP) group mean would be 10 rewards and the mean experimental (ChR2) group would be 20 rewards with a standard deviation of 5. We utilized a power calculator for continuous outcomes of two independent samples, assuming a normal distribution. The result was six samples per group. Each manipulation experiment started with at least six mice included in each group (*Sealed Envelope, 2020*). In each experiment, each animal within a group served as a biological replicate. These studies did not include technical replicates. Masking was not applied during data acquisition but data analyses were semi-automated in MATLAB and performed blind to condition.

## Acknowledgements

These studies were supported by grants DA14241, DA037566, MH077681. LW, DT, and PR were supported by NS022061, MH109104 from the National Institutes of Health, and by the intramural programs of NINDS and NIMH. X-BG was supported by DA046160. RBC was supported by T32-NS007224. This work was funded in part by the State of Connecticut, Department of Mental Health and Addiction Services but this publication does not express the views of the Department of Mental Health and Addiction Services or the State of Connecticut. The views and opinions expressed are those of the authors. We thank Samantha Sheppard for the use of her mouse illustration and animal care assistance and Nadia Jordan-Spasov for genotyping and laboratory help. Li Jiang performed the ex vivo current-clamp recordings. Angela Lee and Wenliang Zhou provided helpful input into experimental planning. Colin Bond, Marcelo Dietrich, Usman Farooq, Onur Iyilikci, Sharif Kronemer, Matthew Pettus, and Zach Saltzman provided insightful discussion and assistance with analysis and figure design. Ralph DiLeone, Stephanie Groman, Hyojung Seo, and Jane Taylor offered helpful discussion about experimental design and analysis. The support teams at Doric Lenses (Alex Côté and

Olivier Dupont-Therrien) and Tucker-Davis Technologies provided discussion, analysis support, and MATLAB code assistance.

# Additional information

## Funding

| Funder | Grant reference number | Author |
|--------|------------------------|--------|
| National Institute on Drug Abuse | DA14241 | Richard B Crouse<br>Kristen Kim<br>Hannah M Batchelor<br>Rufina Kamaletdinova<br>Justin Chan<br>Steven T Pittenger<br>Yann S Mineur<br>Marina R Picciotto |
| National Institute on Drug Abuse | DA037566 | Richard B Crouse<br>Kristen Kim<br>Hannah M Batchelor<br>Rufina Kamaletdinova<br>Justin Chan<br>Steven T Pittenger<br>Yann S Mineur<br>Marina R Picciotto |
| National Institute of Mental Health | MH077681 | Richard B Crouse<br>Kristen Kim<br>Hannah M Batchelor<br>Rufina Kamaletdinova<br>Justin Chan<br>Steven T Pittenger<br>Yann S Mineur<br>Marina R Picciotto |
| National Institute of Neurological Disorders and Stroke | NS022061 | Prithviraj Rajebhosale<br>Lorna W Role<br>David A Talmage |
| National Institute of Mental Health | MH109104 | Prithviraj Rajebhosale<br>Lorna W Role<br>David A Talmage |
| National Institute on Drug Abuse | DA046160 | Xiao-Bing Gao |
| National Institute of Neurological Disorders and Stroke | Intramural | Prithviraj Rajebhosale<br>Lorna W Role |
| National Institute of Mental Health | Intramural | Prithviraj Rajebhosale<br>David A Talmage |
| National Institute of Neurological Disorders and Stroke | T32-NS007224 | Richard B Crouse |

The funders had no role in study design, data collection and interpretation, or the decision to submit the work for publication.

## Author contributions

Richard B Crouse, Conceptualization, Resources, Data curation, Software, Formal analysis, Validation, Investigation, Visualization, Methodology, Writing - original draft; Kristen Kim, Data curation, Formal analysis, Investigation, Methodology, Writing - review and editing; Hannah M Batchelor, Conceptualization, Data curation, Investigation, Methodology, Writing - review and editing; Eric M Girardi, Data curation, Software, Formal analysis, Investigation, Visualization, Writing - review and editing; Rufina Kamaletdinova, Investigation, Methodology, Writing - review and editing; Justin Chan, Data curation, Validation, Investigation, Visualization, Writing - review and editing; Prithviraj Rajebhosale, Lorna W Role, David A Talmage, Conceptualization, Writing - review and editing; Steven T Pittenger, Conceptualization, Investigation, Visualization, Methodology, Writing - review and

editing; Miao Jing, Yulong Li, Resources, Writing - review and editing; Xiao-Bing Gao, Data curation, Formal analysis, Investigation, Writing - review and editing; Yann S Mineur, Conceptualization, Investigation, Methodology, Writing - review and editing; Marina R Picciotto, Conceptualization, Resources, Supervision, Funding acquisition, Project administration, Writing - review and editing

### Author ORCIDs
Richard B Crouse (iD) https://orcid.org/0000-0002-9509-9263
Rufina Kamaletdinova (iD) http://orcid.org/0000-0002-3650-8207
Prithviraj Rajebhosale (iD) http://orcid.org/0000-0001-9893-3025
Lorna W Role (iD) http://orcid.org/0000-0001-5851-212X
Marina R Picciotto (iD) https://orcid.org/0000-0002-4404-1280

### Ethics
Animal experimentation: All procedures were approved by the Yale University Institutional Animal Care & Use Committee in compliance with the National Institute of Health's Guide for the Care and Use of Laboratory Animals. (protocol: 2019-07895).

### Decision letter and Author response
Decision letter https://doi.org/10.7554/eLife.57335.sa1
Author response https://doi.org/10.7554/eLife.57335.sa2

## Additional files

### Supplementary files
• Supplementary file 1. Number of mice that acquired the reward learning behavior, number that were excluded, and any training deviations. (**A**) Mice in the initial BLA ACh3.0 group were trained outside of the sound attenuating chambers. These mice had 5 dof Pre-Training because they were trained concurrently with another cohort of mice (not shown) that required an extra day to reach two consecutive days of 20 rewards earned and were advanced to a VI 20 schedule of reinforcement during Training after 9 d to promote responding. Training was extended to allow all mice to acquire. Due to time constraints during acquisition, Mouse 3 in this cohort was moved to Extinction after 20 d of Training because it had acquired earlier, was earning the most rewards, and we wanted to record more extinction days. (**B**) Mice in the BLA ACh3.0 and NBM-BLA terminal fiber replicate experiments were advanced to 1 d of Extinction after only 7 d of Training due to the COVID-19 shutdown. (**C**) BLA ACh3.0 and NBM-BLA terminal fiber jRCaMP1b mice were analyzed as dual channel mice just through Pre-Training and were instead used as replicates of the BLA ACh3.0 experiment. One of the mice had apparatus errors during Training and had to be excluded. (**D**) Mice in the initial BLA CaMKIIα GCaMP6 were trained outside of the sound attenuating chambers. Mouse 1 progressed from Pre-Training to Training a day earlier than the rest of the group and was able to have an extra day of Training before the 2 d of Extinction. Mice in this group were advanced to a VI 20 schedule of reinforcement during Training after 6–7 d to promote responding. Training was extended to allow more mice to acquire.

• Transparent reporting form

### Data availability
All data generated or analysed during this study are included in the manuscript and supporting files. Source data files have been provided for all experiments on Dryad Digital Repository: https://doi.org/10.5061/dryad.3xsj3txcf.

The following dataset was generated:

| Author(s) | Year | Dataset title | Dataset URL | Database and Identifier |
|---|---|---|---|---|
| Picciotto M, Crouse RB, Kim K, Batch- | 2020 | Acetylcholine is released in the basolateral amygdala in response | https://doi.org/10.5061/dryad.3xsj3txcf | Dryad Digital Repository, 10.5061/ |

elor H, Kamaletdi-
nova R, Chan J,
Rajebhosale P, Pit-
tenger S, Role L,
Talmage D, Jing M,
Li Y, Gao X-B,
Mineur Y

to predictors of reward and
enhances learning of cue-reward
contingency

dryad.3xsj3txcf

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

# Appendix 1

## Supplemental methods

### Ex Vivo electrophysiology

#### Slice preparation

Coronal brain slices were prepared from virus injected mice after 3 weeks from surgery. Animals were anesthetized with a mixture of ketamine and xylazine (100 mg ketamine and 6 mg xylazine/kg body weight injected ip). Then the mice were transcardially perfused with a sucrose-based solution (see below). After decapitation, the brain was rapidly transferred into a sucrose-based cutting solution bubbled with 95% $O_2$ and 5% $CO_2$ and maintained at ~3°C. This solution contained (in mM): sucrose 230; KCl 2.5; $MgSO_4$ 10; $CaCl_2$ 0.5; $NaH_2PO_4$ 1.25; $NaHCO_3$ 26; glucose 10 and pyruvate 1.5. Coronal brain slices (300 μm) were prepared using a Leica VT1000S vibratome (Leica Biosystems Inc). Slices were equilibrated with a mixture of oxygenated artificial cerebrospinal fluid (aCSF) and sucrose-based cutting solution at room temperature (24–26°C) for at least 1 hr before transfer to the recording chamber. Pyruvate (0.15–0.75 mM) was added to reduce oxidative damage and enhance survival. With this protocol, slices are initially incubated in a mixture of 50% cutting solution with pyruvate and 50% aCSF (in mM): sucrose 115; NaCl 63; KCl 2.5; $NaH_2PO_4$ 1.25; $MgSO_4$ 5; $CaCl_2$ 1.25; $MgCl_2$ 1; $NaHCO_3$ 26; glucose 10; and sodium pyruvate 0.75 at 35°C for 30 min and then transferred to a mixture of 10% cutting solution and 90% aCSF (in mM): sucrose 23; NaCl 113.4; KCl 2.5; $NaH_2PO_4$ 1.25; $MgSO_4$ 1; $CaCl_2$ 1.85; $MgCl_2$ 1.8; $NaHCO_3$ 26; glucose 10; and sodium pyruvate 0.15 at 35°C for 1–4 hr before recording. The slices were continuously superfused with aCSF at a rate of 2 ml/min containing (in mM); NaCl 126, KCl 2.5, $NaH_2PO_4$ 1.25, $NaHCO_3$ 26, $CaCl_2$ 2, $MgCl_2$ 2 and glucose 10 bubbled with 95% $O_2$ and 5% $CO_2$ at room temperature.

### Electrophysiological recordings

Brain slices were placed on the stage of an upright, infrared-differential interference contrast microscope (Olympus BX51WI, Olympus). NBM neurons were visualized with a 40 × water-immersion objective by infrared microscopy (COHU 4915 camera, COHU, Inc, Poway, CA). Patch electrodes with a resistance of 4–6 MΩ were pulled with a laser-based micropipette puller (P-2000, Sutter Instrument Company). Signals were recorded with a Multi Clamp 700A amplifier and pClamp10 software (Molecular Devices, Inc, San Jose, CA). The pipette solution contained (in mM) 130 K-gluconate, 2 KCl, 2 $MgCl_2$, 10 HEPES, 0.5 EGTA, 1 ATP and 0.2 GTP (pH = 7.3).

To examine action potential firing frequency, NBM neurons were recorded in a current clamp configuration after forming a giga-ohm seal. Membrane potentials were clamped at −60 mV by injecting 0- ~ 50 pA current through the recording electrode as needed. Cells that maintained steady membrane potentials for at least five mins were included in the analysis.

### Optogenetic stimulation ex vivo

Channelrhodopsin was activated with a train of light flashes delivered through the 40× microscope objective. The light source was an Olympus x-cite 120Q lamp (Olympus) gated with a TTL controlled shutter (LAMBDA SC, Sutter Instrument). The filter cube contained an HQ480/40x excitation filter, a Q505lp bypass filter and an HQ535/50 m emission filter (Chroma Technology Corp., Bellows Falls, VT). The fluorescence illumination intensity delivered at the brain slices was adjusted to 1–3 mW/$mm^2$, measured with a PM100D optical power and energy meter (Thorlabs Inc, Newton, NJ). In the NBM, cholinergic neurons were identified by EGFP fluorescence and light flashes were delivered at 1 Hz, 5 Hz, 10 Hz, 15 Hz, 20 Hz, 25 Hz, and 30 Hz.

### Cued self-stimulation

After xtinction, responding was reinstated in Training for 2 d. Then mice underwent a modified Training paradigm where correct nose pokes yielded only laser stimulation, without Ensure delivery.

## Real time place preference

An empty, clear mouse cage (29.5 cm × 19 cm × 12.5 cm) had half of its floor covered in printer paper to provide a distinct floor texture. A video camera was placed above the cage and was connected to a computer running EthoVision XT (version 10.1.856, Noldus, Wageningen, Netherlands) to track the position of the mouse and deliver optical stimulation when the mouse was on the laser-paired side (via TTL pulse to OTPG_4 laser controller (Doric Lenses) connected to the laser; 20 Hz, 25 ms pulses). Mice were randomly assigned and counterbalanced to receive laser stimulation only on one side of the cage. Mice were allowed free access to either side for 15 min during a session. Baseline was established in the absence of optical stimulation on day 1. Mice then received optical stimulation on Day two only when on the laser-paired side. Data are presented as percent time spent on the laser-paired side.

## Progressive ratio testing

In the progressive ratio test, mice were given 60 min to nose poke for ensure and 2 s of optical stimulation on a progressive ratio schedule (escalations given below). Training day escalation: 1, 2, 2, 2, 2, 3, 3, 3, 3, 3, 5, 5, 5, 5, 5, 8, 8, 8, 8, 8, 8, 11, 11, 11, 11, 11, 11, 15, 15, 15, 15, 15, 15, 22, 22, 22, 22, 22, 33, 33, 33, 33, 33, 44, 44, 44, 44, 44, 55, 66, 77, 88, 99, 133, 166, 199, 255, 313, 399, 499, 599, 777, 900,1222. Test Day escalation: 1, 2, 4, 6, 9, 12, 15, 20, 25, 32, 40, 50, 62, 77, 95, 118, 145, 178, 219, 268, 328, 402, 492, 603, 777, 900, 1222.

## Locomotor activity
### Optical stimulation

Mice were placed in a square box (47 cm × 47 cm × 21 cm) for 20 min with a floor of filter paper that was changed between mice. During the 3$^{rd}$ 5 min bin of the session, mice received optical stimulation (20 s on/off, 20 Hz, 25 ms pulses). Locomotor activity was recorded via overhead camera and analyzed in 5 min bins with EthoVision.

### Antagonists

Locomotor data was collected using an Accuscan Instruments (Columbus, Ohio) behavior monitoring system and software. Mice were individually tested in empty cages, with bedding and nesting material removed to prevent obstruction of infrared beams. Mice were injected (i.p.) with saline, mecamylamine (1 mg/kg, Sigma), scopolamine (0.5 mg/kg, Sigma), or mecamylamine+scopolamine (1 mg/kg and 0.5 mg/kg, respectively) 30 min before locomotor testing. Locomotion was monitored for 20 min using 13 photocells placed 4 cm apart to obtain an ambulatory activity count, consisting of the number of beam breaks recorded during a period of ambulatory activity (linear motion rather than quick, repetitive beam breaks associated with behaviors such as scratching and grooming).

## Light/dark box exploration

A rectangular box was divided evenly into a light (clear top, illuminated by an 8W tube light) and dark (black walls, black top) side with a black walled divider in the middle with a small door. The lid and divider were modified to allow the optical fiber and patch cord to pass through freely. Mice were placed facing the corner on the light side furthest from the divider and the latency to crossing to the dark side was measured. The number of crosses and time spent on each side were measured for 6 min following the initial cross.

