## [Decision Letter]

**Acceptance summary:**

The reviewers agree that this research is novel, intriguing and important, as well as technically accomplished, set of findings and that the authors have sufficiently addressed all the comments they raised.

**Decision letter after peer review:**

Thank you for submitting your article "Acetylcholine is released in the BLA in response to predictors of reward and enhances learning of cue-reward contingency" for consideration by *eLife*. Your article has been reviewed by two peer reviewers, and the evaluation has been overseen by a Reviewing Editor and Laura Colgin as the Senior Editor. The following individuals involved in review of your submission have agreed to reveal their identity: David M Lovinger (Reviewer #1); Gavan P McNally (Reviewer #2).

The reviewers have discussed the reviews with one another and the Reviewing Editor has drafted this decision to help you prepare a revised submission.

Summary:

The manuscript by Crouse et al. uses a variety of approaches to provide important new information about the role of acetylcholine release (ACh) in the basolateral amygdala (BLA) in appetitive learning. The authors use a variety of powerful techniques to examine and manipulate ACh levels and neuronal activity in this brain region. The findings support the conclusion that ACh facilitates conditioning in a cue-driven nosepoke task through actions on a muscarinic receptor. The quality of the data are generally high, especially the fiber photometry data that exhibits little-to-no artifactual signal in most of the experiments. Important controls are included for the most crucial experiments. There are some important and real design strengths here. These include inclusion of data from a pretraining phase for each experiment (showing neatly that the ACh system is neither non-specifically recruited nor non-specifically contributing to general behaviours in this task), concurrent measurement of release via ACh3.0 and terminal activity via jRCaMP1b (showing strong temporal agreement between the measures), pharmacological identification of the critical receptor family, augmentation of learning in the task via ChR2 excitation, and inclusion of useful controls (non-contingent stimulation) that constrain interpretation. The manuscript is well-written with clear findings. Overall, the findings reported here are an important advance and are interesting for their knowledge gains about both ACh and amygdala function in a fundamentally important learning process. There are a few aspects of the study that require clarification, and additional analysis of existing data and other information should be included. No new experiments are required.

Essential revisions:

1) The first revision relates to the unpaired stimulation experiment. There are three points to address here:

– The duration of the ACh effect suggested by the explicitly unpaired stimulation procedure appears to be on the order or 10s of seconds. This is consistent with mAChR involvement as the authors suggest. It is possible that ACh release induced by optogenetic stimulation is longer-lasting than that induced by CS delivery. This possibility should be considered, and the authors might want to discuss the known durations of ACh modulatory actions. The possibility of co-release of another neurotransmitter or modulator with this optical stimulation should also be considered (Saunders et al., 2015). It may be that strong ACh release activates long-term synaptic plasticity independent of neural responses to CS or reward delivery, and the authors might also mention this possibility.

– How did the authors determine the pattern of light stimulation used in the optogenetic experiments? Were effects of this stimulation pattern examined in slices (Figure 4B only shows responses to lower frequency stimulation)? Please state the frequency and duration of ChR2 stimulation in the main text (subsection “Stimulation of cholinergic terminals in BLA improves cue-reward learning”, first paragraph).

– One of the surprising findings here was that yoked ChR2 stimulation was as effective in enhancing performance. This was surprising given how well timed the endogenous signals are in this task. This finding was not really explored in the Discussion and it appears to contradict interpretation at some key places in the Discussion (e.g., –subsection “Increasing BLA acetylcholine levels enhances cue-reward learning”). Also, the yoked stimulation paradigm may actually be something of a "semi-yoking" procedure in that stimulation was not time-locked to stimulation in the original training group. Have other authors used the term yoked for this exact type of procedure?

2) It is surprising that such a complicated conditioning paradigm (i.e. presentation of a time-out period after incorrect responses in addition to reward for correct responses) is required for learning to respond specifically to the reward-predicting CS. Perhaps this is related to the several days of shaping as this may reinforce a strategy of near-automatized nosepoking that is then harder to modify in subsequent conditioning phases. Was it necessary to shape mice for this long? Also, the task appears to use a Pavlovian-to-instrumental transfer (PIT) conditioning design. If this is the case, perhaps the authors should use that term to describe the paradigm at some point in the manuscript, as that is a generally-accepted term that will be instantly recognized by many readers.

Following the comment above, it isn't clear if all mice learned the task, both in the training period and at the end (i.e. were levels of nose poking greater for correct versus incorrect?). The separation of these data between main and supplementary figures, and the required use of different scales, makes it difficult for the reader to understand the comparative frequencies of the behaviours. The authors should indicate what proportion of mice met the criteria for learning and a clearer statement of the criteria for successful completion of each training phase should also be included in the Materials and methods.

3) There are a few points with the photometry data that need to be addressed.

– In the photometry experiments, the group sizes are relatively small (4) and on some important control experiments (cross talk between the sensors) only single subject data are shown, and this might be addressed in the manuscript. The absence of any quantitative analysis of photometry is a also potential issue, but may well be related to sample sizes. The authors do provide a thoughtful and careful descriptive, qualitative analysis of photometry data, but statistical analyses (either using trial or subjects) appear absent. For the reader, this is especially relevant when trying to understand the relevance of signals around the incorrect nosepoke. There are approaches (within-subject bootstrapped confidence intervals to ask if CI includes 0% df/f or permutation tests) that could be applied. Again, the most straightforward and transparent solution could be to explain this choice to readers.

– For the fiber photometry data, how were data handled if there was mirroring in the 405 channel? If the data in this channel do not reflect calcium changes, presumably the data in the other channels are most likely artifacts. Were these periods then eliminated from analysis as others have done?

– Figure 3—figure supplement 1C is the only instance where the 405 channel shows a noticeable change. Do the authors have any idea why the large decrease was observed in this experiment?

---

## [Author Response]

Essential revisions:1) The first revision relates to the unpaired stimulation experiment. There are three points to address here:– The duration of the ACh effect suggested by the explicitly unpaired stimulation procedure appears to be on the order or 10s of seconds. This is consistent with mAChR involvement as the authors suggest. It is possible that ACh release induced by optogenetic stimulation is longer-lasting than that induced by CS delivery. This possibility should be considered, and the authors might want to discuss the known durations of ACh modulatory actions.

We agree with the reviewers that optogenetic-mediated release of ACh could be longer lasting than that induced by tone onset (CS delivery, or in early phases of training, correct nosepoke). We have added an explicit statement of this:

“It is also possible that optogenetic-mediated ACh release may last longer than endogenous, tone-evoked release. A simultaneous stimulation and recording approach would be required to compare ACh release under both conditions (Pisansky et al., 2019).”

We also added text to the Introduction that explicitly mentions the differing timescales of nAChR and mAChR action:

“The effect of ACh signaling can differ depending on the receptor, as metabotropic mAChRs work on a slower timescale than the rapid, ionotropic nAChRs (Gu and Yakel, 2011; Picciotto et al., 2012).”

The possibility of co-release of another neurotransmitter or modulator with this optical stimulation should also be considered (Saunders et al., 2015).

We agree that co-release of other neurotransmitters along with acetylcholine should be considered and unintentionally neglected to discuss this in the previous version of the manuscript. We have added the following text to the Discussion:

“It is important to note that basal forebrain neurons have been demonstrated to co-release ACh and GABA (Ma et al., 2018; Saunders et al., 2015), and cholinergic basal forebrain neurons that project to the BLA have been shown to co-express a glutamate transporter (Ma et al., 2018; Poulin et al., 2006). […] Future studies employing additional fluorescent neurotransmitter sensors (Marvin et al., 2013, 2018, 2019) could help understand the interaction between the different signals employed by basal forebrain neurons.”

It may be that strong ACh release activates long-term synaptic plasticity independent of neural responses to CS or reward delivery, and the authors might also mention this possibility.

We agree this is possible and indeed, is likely. We believe that the neural signals induced by CS or reward are likely necessary to induce firing of basal forebrain cholinergic neurons, but once ACh is released, it likely influences the circuit for tens of seconds and induces plasticity that alters BLA responses on a much longer timescale. This is addressed in the Discussion:

“This suggests that the effect of increased cholinergic signaling in the BLA is long lasting, and that stimulation during a learning session is sufficient to potentiate synaptic events linking the cue to a salient outcome – even if CS and/or reward delivery are presented tens of seconds later. […] Coupled with pharmacological evidence demonstrating that muscarinic signaling is necessary for reward learning in this task, these results suggest the involvement of metabotropic signaling downstream of muscarinic receptors that outlasts the initial cholinergic stimulation.”

– How did the authors determine the pattern of light stimulation used in the optogenetic experiments?

We have elaborated on how we determined the pattern of light stimulation in the Materials and methods:

“The stimulation pattern was 25 ms pulses at 20 Hz for 2 sec modified from parameters in (Jiang et al., 2016). […] A 25 ms pulse width was used because our lasers were not able to generate sufficient power with 5 ms pulses.”

Were effects of this stimulation pattern examined in slices (Figure 4B only shows responses to lower frequency stimulation)?

We have now recorded from NBM ChAT^+^ cells expressing ChR2 in slices at different stimulation frequencies. It appears that all frequencies elicited time locked depolarization in ChAT neurons and action potentials were apparent in slices up to ~10 Hz stimulation. These data are now included in Figure 4—figure supplement 2.

Please state the frequency and duration of ChR2 stimulation in the main text (subsection “Stimulation of cholinergic terminals in BLA improves cue-reward learning”, first paragraph).

We have added the frequency and duration of ChR2 stimulation to these lines:

“After operant familiarization, ChAT+ NBM-BLA terminals were stimulated via bilateral optical fibers (2 sec, 20 Hz, 25 ms pulses) triggered by a correct nose poke throughout both Pre-Training (Figure 4C) and Training (Figure 4D).”

– One of the surprising findings here was that yoked ChR2 stimulation was as effective in enhancing performance. This was surprising given how well timed the endogenous signals are in this task. This finding was not really explored in the Discussion and it appears to contradict interpretation at some key places in the Discussion (e.g., –subsection “Increasing BLA acetylcholine levels enhances cue-reward learning”).

We were also quite surprised by this finding precisely because of the tight time-locking seen in the fiber photometry experiments. We have now expanded the Discussion of the outcome of the non-contingent experiment and more explicitly discuss this contradiction:

“The timing of cholinergic signaling can be a critical factor in the induction of synaptic plasticity in other brain regions, so we hypothesized that the enhancement of cue-reward learning observed might be dependent upon when NBM-BLA terminal fibers were stimulated with respect to tone presentation and/or behavioral responses (Gu and Yakel, 2011). […] Coupled with pharmacological evidence demonstrating that muscarinic signaling is necessary for reward learning in this task, these results suggest the involvement of metabotropic signaling downstream of muscarinic receptors that outlasts the initial cholinergic stimulation.”

Also, the yoked stimulation paradigm may actually be something of a "semi-yoking" procedure in that stimulation was not time-locked to stimulation in the original training group. Have other authors used the term yoked for this exact type of procedure?

We agree that our paradigm is actually closer to “semi-yoking” with the number of stimulations matched to the concurrently running ChR2 counterpart. To avoid confusion, we changed the name of the “Yoked” group to “Non-contingent ChR2” throughout the manuscript and modified the description of the group:

“…we repeated the experiment in an independent cohort of mice with an additional non-contingent ChR2 group that received the same number of stimulation trains as the contingent ChR2 group, but in which light stimulation was explicitly unpaired with task events (Figure 6A + Figure 6—figure supplement 1).”

2) It is surprising that such a complicated conditioning paradigm (i.e. presentation of a time-out period after incorrect responses in addition to reward for correct responses) is required for learning to respond specifically to the reward-predicting CS. Perhaps this is related to the several days of shaping as this may reinforce a strategy of near-automatized nosepoking that is then harder to modify in subsequent conditioning phases. Was it necessary to shape mice for this long?

We were also surprised that a timeout period was required for mice to acquire the contingency. However, in our initial piloting for the behavioral paradigm, mice did not decrease incorrect nose poking after 7 days of Pre-Training, and only decreased incorrect nose poking after including the timeout period. We agree that this could be related to the several days of behavioral shaping before training. The one day of reward receptacle “Free Reward” and 4-5 days of nose poke “FR1 Shaping” were required to ensure all mice in a cohort earned at least 20 rewards for two consecutive days before advancing to Pre-Training. We preferred to advance all mice together after 4-5 days of shaping in order to avoid excluding mice from the experiment. This also ensured that each mouse underwent the same number of training sessions within and across experiments. This was not always possible, though, because in the initial BLA ACh3.0 and BLA CaMKIIα fiber photometry experiments we trained mice outside of the sound attenuating chambers (Figure 2A-E + Figure 3) and required one extra day of FR1 shaping to learn the task. We think it could be interesting to investigate if a timeout period is necessary to learn the cue-reward contingency if mice are advanced individually based on their behavior during shaping, however, for the purposes of these experiments, we chose to prioritize consistency in training across cohorts. (Note: in the revised manuscript we have removed the term “shaping” and changed it to “operant familiarization” to avoid confusion. See point below.)

Also, the task appears to use a Pavlovian-to-instrumental transfer (PIT) conditioning design. If this is the case, perhaps the authors should use that term to describe the paradigm at some point in the manuscript, as that is a generally-accepted term that will be instantly recognized by many readers.

We understand the reviewer’s concern, but we believe there may be a misunderstanding here because of our use of the term “shaping”, which could be confused with the autoshaping commonly seen in PIT. However, the “shaping” performed here was simply familiarization with the operant chamber and the nosepoke response to encourage mice to learn to perform the operant task. To avoid confusion, we now refer to this phase as “operant familiarization”. Mice are not exposed to the auditory tone signaling that an active nose poke can be made to receive a reward until the “Pre-Training” phase of the task. More directly, the mice do not form a Pavlovian association between the tone and reward before being required to nose poke for reward (Cartoni et al., 2016).

Following the comment above, it isn't clear if all mice learned the task, both in the training period and at the end (i.e. were levels of nose poking greater for correct versus incorrect?).

All mice included in fiber photometry figures (Figures 1-3) acquired the task. Not all mice in the optogenetic or antagonist studies reached the criterion for acquisition, however, and the proportion of animals that reached the 20-reward threshold or were excluded is now listed in Supplementary file 1.

The separation of these data between main and supplementary figures, and the required use of different scales, makes it difficult for the reader to understand the comparative frequencies of the behaviours.

We agree that separation of rewards earned and incorrect nose pokes between main and supplementary figures (in the case of the fiber photometry experiments) makes it difficult to compare the frequencies of correct and incorrect responding. We did not include the rewards earned and incorrect nose pokes overlaid in the same graph because the far greater number of incorrect nose pokes during the “Pre-Training” period dwarfs the rewards earned and prevents the overlay from being useful. However, we provide the data used in the figures in a readily plottable format (Excel sheets in Dryad).

The authors should indicate what proportion of mice met the criteria for learning and a clearer statement of the criteria for successful completion of each training phase should also be included in the Materials and methods.

In general, mice progressed from FR1 Nosepoke Training to Pre-Training after 30 rewards were earned by the group for at least two consecutive days (which was when the majority of mice in the group passed the individual threshold for acquisition of 20 rewards earned), typically 4 days. Mice progressed from Pre-Training typically after 4 days or when all mice had earned at least 20 rewards for two consecutive days, whichever came first. Task acquisition for individual animals was defined as earning 20 rewards during the Training phase of the task. Training lasted for 12 days when possible, but was extended for fiber photometry experiments carried outside of the sound attenuating chambers.

For the optogenetic stimulation and pharmacological antagonist studies, some groups included mice that never acquired the task by the end of Training. All of these mice underwent the standard 4 days of Pre-Training and 12 days of Training (as determined by initial task piloting). All mice shown in the fiber photometry figures acquired the task, but some mice in these cohorts did not acquire the task by the end of Training and were excluded from analyses We now include Supplementary file 1 to demonstrate more clearly the proportion of mice that met acquisition criteria, any excluded mice, and any deviations from the standard paradigm.

We have defined task acquisition and criteria more clearly for successful completion of each training phase in the Materials and methods section as follows:

“After the mice reached criterion on FR1 operant familiarization (group average of 30 rewards for 2 consecutive days, usually 4-5 days), mice were advanced to the Pre-Training phase. […] Mice were considered to have acquired the task after earning 20 rewards during the Training phase of the task.”

3) There are a few points with the photometry data that need to be addressed.– In the photometry experiments, the group sizes are relatively small (4) and on some important control experiments (cross talk between the sensors) only single subject data are shown, and this might be addressed in the manuscript.

We were also concerned about the group sizes of the photometry experiments, so we have added replication groups for the BLA ACh3.0 (4 mice), NBM-BLA ChAT^+^ terminal fiber GCaMP7s (2 mice), and BLA CaMKII^+^ GCaMP6s (4 mice) experiments. The BLA ACh3.0 and BLA CaMKII^+^ GCaMP6s replication cohorts cannot be easily combined with the original cohorts due to differences in the number of behavioral sessions performed, but importantly, the replication studies show comparable outcomes as the original experiments and now all groups have at least one cohort that was trained inside the sound attenuating chambers in the optimized behavioral setup. Since the original and NBM-BLA ChAT^+^ terminal fiber GCaMP7s cohorts acquired at similar rates, they were combined to allow for cohort bootstrapping. The new BLA ACh3.0 cohort underwent the standard 4 days of Pre-Training, 12 days of Training, and 3 days of Extinction training, while the new NBM-BLA ChAT^+^ terminal fiber GCaMP7s and BLA CaMKII^+^ GCaMP6s cohorts ended Training prematurely at 7 days followed by 1 day of Extinction due to the COVID-19 shutdown. Data for all replication cohorts are shown in new supplementary figures, and mouse numbering is continued from original cohorts to avoid confusion.

The sensor cross talk control experiment was a proof of principle with only a single animal so we do not feel justified in drawing strong conclusions from the experiment. We did not refer to it in the Discussion, but wanted to include it as a supplementary figure, since it was a striking example of convergence that we consider as one of many control experiments. We have modified our description of the results from this one animal to further caution readers from drawing overly strong conclusions from the data as follows:

“While this was only a single animal and proof of principle for future studies, we found that NBM-BLA cholinergic terminal activity coincided with ACh levels (Figure 2—figure supplement 9F-G). Importantly, this relationship between ACh release and NBM-BLA terminal fiber activity was not explained by signal crosstalk (Figure 2—figure supplement 9H-I), further indicating that the BLA ACh measured comes at least in part from the NBM.”

The absence of any quantitative analysis of photometry is a also potential issue, but may well be related to sample sizes. The authors do provide a thoughtful and careful descriptive, qualitative analysis of photometry data, but statistical analyses (either using trial or subjects) appear absent. For the reader, this is especially relevant when trying to understand the relevance of signals around the incorrect nosepoke. There are approaches (within-subject bootstrapped confidence intervals to ask if CI includes 0% df/f or permutation tests) that could be applied. Again, the most straightforward and transparent solution could be to explain this choice to readers.

We were hesitant to perform quantitative analyses because of the low sample sizes, as the reviewers inferred, and initially decided to limit ourselves to qualitative descriptions. We thank the reviewers for the suggestion of bootstrapped confidence intervals and have incorporated both within- and across- subject bootstrapped confidence intervals (bCI) to assist readers in interpreting our results (Jean-Richard-dit-Bressel et al., 2020). We have modified our line plots with FP data to incorporate significance bars to denote time bins when bCIs do not contain Z%ΔF/F0 = 0 for at least 0.5 sec. We have also included new bCI plots that show similar data to the tone-poke-reward retrieval heatmaps with indicators of significance. The same subset of days used for the across-mouse averaged data heatmaps was used for an example mouse (within mouse bCI) and averaged (when possible) bCI plots.

– For the fiber photometry data, how were data handled if there was mirroring in the 405 channel? If the data in this channel do not reflect calcium changes, presumably the data in the other channels are most likely artifacts. Were these periods then eliminated from analysis as others have done?

We did not observe gross systematic artefacts or instances when the 405 channel mirrored the 465 channel’s transients. In order to minimize the processing of our data we opted to run the analyses for the reference and signal channels separately instead of using a “reference-corrected” ΔF/F0. In addition to the non-Z-scored % ΔF/F0 comparison of reference and signal channels for correct and incorrect nose pokes, we ran the entire analysis pipeline for the reference channels for each experiment and did not see anywhere near the same magnitude of signal. For the case of the BLA CaMKIIα GCaMP6s signal, mirroring was observed due to a known phenomenon as the sensor binds to calcium (see response to the point below). We did not include a duplication of all graphs to avoid doubling the number of figures, but our Dryad upload includes an option for readers to easily rerun the entire analyses for the reference channel and compare the outputs themselves. We include one example comparison as Author response image 1.

In addition, to avoid confusion, we now explicitly define mirroring in the Materials and methods section to refer only to the situation when the reference channel moves in the opposite direction to the signal channel (see Point below).

– Figure 3—figure supplement 1C is the only instance where the 405 channel shows a noticeable change. Do the authors have any idea why the large decrease was observed in this experiment?

Based on published studies (see citations below), we believe that the mirroring in the 405 channel observed in the BLA CaMKIIα GCaMP experiment is due to a decrease in calcium-unbound GCaMP (as calcium-bound GCaMP increases). This is due to a reference LED that is not at the “true” isosbestic point for GCaMP. We have explained this more clearly:

“For the BLA CaMKIIα cell recordings (Figure 3—figure supplement 1C-D), the reference channel displayed some mirroring (moving in the opposite direction) compared to the signal. This is likely because 405 nm is not the “true” isosbestic point for GCaMP and we were instead measuring some changes in calcium-unbound GCaMP rather than calcium-insensitive GCaMP signal alone (Barnett et al., 2017; Kim et al., 2016; Sych et al., 2019).”

Reference:

Cartoni, E., Balleine, B., & Baldassarre, G. (2016). Appetitive Pavlovian-instrumental Transfer: A review. Neuroscience and Biobehavioral Reviews, 71, 829–848. https://doi.org/10.1016/j.neubiorev.2016.09.020